# Growing Transformers: Modular Composition and Layer-wise Expansion on a Frozen Substrate

## Abstract

Monolithic end-to-end pretraining can become memory-bounded because optimizer states must be stored for all trainable parameters, limiting scaling under a fixed accelerator memory budget. We study a constructive alternative: grow a decoder-only Transformer by stacking new blocks and training only the newly added top block(s) (plus the LM head) while keeping embeddings and previously trained blocks frozen. For larger runs we optionally insert LoRA phases to enable limited global readjustment under the same trainable-parameter budget.

In a large-scale feasibility study, we grow a 2.3B-parameter model while keeping the number of trainable parameters (and thus optimizer-state footprint) bounded at 740M per stage, fitting optimizer state into a single 80GB H100 and enabling data-parallel training without optimizer-state sharding. In a controlled 9-layer (248M) study, layer-wise growth on a frozen Unicode substrate achieves comparable downstream performance to monolithic end-to-end training at the same final architecture, while reducing peak trainable parameters from 248M (monolithic) to 105M (constructive, peak stage). We also show that growth remains viable with extremely low-capacity 16-dim binary frozen embeddings (n_embed=16).

Finally, in an extended 68.9B-token run on FineWeb-Edu + Cosmopedia, we grow a 16-layer (269.7M) model with frozen 16-dimensional binary token-ID embeddings and a fixed 117.5M trainable-parameter budget, reaching 28.92% MMLU and 19.10% SQuAD EM after an interleaved global LoRA stage.

Overall, layer-wise growth on a frozen substrate

provides a practical option for memory-bounded training, incremental scaling, and controlled studies of depth-driven capability emergence.

## 1. INTRODUCTION

Scaling language models is commonly done via monolithic end-to-end pretraining of a fixed architecture. This paradigm is effective but often memory-bounded due to optimizer states and activation storage, and it offers limited flexibility for incremental scaling and controlled studies of capability emergence.

The embedding layer is fixed throughout training using the deterministic mapping of Anonymous (2025). Results reported there indicate that Transformers can learn useful representations even with a frozen, non-semantic embedding layer, supporting the hypothesis that a stable input substrate can enable meaningful computation to be built incrementally in deeper layers. The work then focuses on: (i) progressive depth growth with lower blocks kept frozen, (ii) training under an explicit fixed budget on trainable parameters (optimizer state), (iii) scaling to 2.3B parameters under a single-device optimizer-state constraint, and (iv) controlled comparisons, including monolithic training on the same frozen substrate.

This paper asks: can we incrementally grow depth by training only newly added layers while maintaining stable training and competitive downstream performance relative to monolithic training of the same final architecture?

To validate this hypothesis, we conduct three sets of experiments:

- **Feasibility under a fixed trainable-parameter budget:** we grow a Transformer from 1 to 6 layers while keeping the number of trainable parameters (and thus the optimizer-state footprint) approximately constant (740M). This provides a practical recipe for training models whose full end-to-end optimizer state would not fit on a given budget (e.g. our case fitting on a single H100 GPU). The training throughput was scaled using data parallelism across additional accelerators. This experiment shows how complex capabilities emerge as

---

[1]Anonymous Institution, Anonymous City, Anonymous Region, Anonymous Country. Correspondence to: Anonymous Author <anon.email@domain.com>.

the model grows from 1 to 6 layers, demonstrating a practical path to training models that would otherwise be infeasible to train monolithically on such hardware.

- **Controlled Comparative Study:** Second, to rigorously evaluate the performance of our approach, we conduct a controlled experiment on smaller (247.6M), 9-layer models. We compare our constructive method using frozen embeddings against three critical baselines: (a) a traditional, monolithically trained model of the same final architecture; (b) a constructively grown model using standard trainable embeddings; and (c) an ablation with a minimalist embedding composed of 16-component binary vectors to test the limits of the principle. We focus on comparability and task-dependent trade-offs, rather than claiming universal improvements.

- **Web-scale scaling with interleaved global LoRA under the same budget:** We grow a 16-layer (269.7M) model using frozen 16-dimensional binary token-ID embeddings on FineWeb-Edu + Cosmopedia for 68.9B tokens. The trainable set is kept fixed at $\approx 117.5$M (newest 4 blocks + LM head), and we interleave global LoRA stages that reallocate the same trainable budget to adapters across all layers. This reaches 28.92% MMLU and 19.10% SQuAD EM at the final 1-16 LoRA stage.

## 1.1. Contributions.

We provide:

- (1) a simple layer-wise growth procedure for decoder-only Transformers that operates under a fixed trainable-parameter (optimizer-memory) budget;

- (2) a large-scale feasibility demonstration up to 2.3B parameters showing stable optimization when monolithic optimizer state would be memory-prohibitive on a single device;

- (3) a controlled comparison against monolithic baselines, including monolithic training on the same frozen substrate, to disentangle growth effects from embedding effects;

- (4) an ablation showing viability even with 16-dimensional binary frozen token-ID embeddings;

- (5) an extended 68.9B-token run on FineWeb-Edu + Cosmopedia demonstrating that constructive growth with frozen 16-dimensional binary token IDs scales to 16 layers (269.7M base) and that interleaved global LoRA phases can substantially improve downstream metrics under the same fixed trainable-parameter budget.

## 2. RELATED WORK

Our work builds upon several lines of research in neural network training and modularity, but offers a distinct synthesis enabled by our frozen-embedding foundation.

### 2.1. Greedy Layer-Wise Training

The idea of training deep networks one layer at a time was pioneered by Hinton et al. (Hinton et al., 2006) for Deep Belief Nets and explored by Bengio et al. (Bengio et al., 2007) for deep autoencoders. These methods were primarily used for pre-training to find a good initialization for subsequent end-to-end fine-tuning. Our approach differs in emphasis: layer-wise growth is used as the primary training mechanism under a fixed trainable-parameter budget, rather than merely as initialization for full end-to-end training.

### 2.2. Progressive and Modular Architectures

Progressive Neural Networks (PNNs) (Rusu et al., 2016) achieve continual learning by adding new network "columns" for each new task. Adapter-based methods like AdapterFusion (Pfeiffer et al., 2021) inject small, trainable modules into a frozen base model. Our layer-wise growth is a form of vertical, rather than horizontal, expansion. The use of Low-Rank Adaptation (LoRA) (Hu et al., 2022) in our growth stages serves as an efficient tool for global readjustment (Schulman & Lab, 2025), but the core principle remains constructive and layer-wise.

Large-scale training is often limited by optimizer-state memory, which motivates sharded optimizers and distributed training systems (e.g., ZeRO/FSDP-style approaches), offloading, and low-precision optimizers. Our method is complementary: instead of sharding optimizer states, we reduce their footprint by constraining the number of trainable parameters per stage (new blocks + LM head), enabling simple data-parallel training without optimizer-state sharding in our setting.

In contrast to prior studies of progressive depth growth, we explicitly evaluate the interaction between (a) a frozen, non-semantic embedding substrate and (b) strict freezing of previously trained layers, and we include monolithic baselines with the same frozen substrate to disentangle these effects.

### 2.3. Progressive Layer-Wise Growth

Instead of initializing a deep, N-layer Transformer and training it all at once, we "grow" it iteratively.

1. **Initialization:** Create a model $M_1$ with a single Transformer block ($n_{\text{layer}} = 1$) on top of the frozen embed-

ding layer.

2. **Train Layer 1:** Train $M_1$ on the full corpus until convergence.

3. **Freeze and Stack:** Freeze the weights of the trained block in $M_1$. Add a new, randomly initialized Transformer block on top, creating model $M_2$.

4. **Train Layer 2:** Train $M_2$ with the full forward pass through all layers; only the parameters of the newly added Transformer block and the output projection head are updated; all embeddings and previously trained blocks remain frozen.

5. **Iterate:** Repeat this process, adding either a single Transformer block or a small block group (e.g., 3 blocks in our controlled study) on top, and training only the newly added blocks (and the output head) while keeping embeddings and previously trained blocks frozen.

This layer-wise process can be strategically interspersed with holistic fine-tuning stages. For deeper models (e.g., after reaching 3 or 5 layers), we introduce an alternative training phase. Instead of adding a new layer, we freeze the entire stack and apply Low-Rank Adaptation (LoRA) (Hu et al., 2022) adapters to all existing layers.

## 2.4. Frozen 16-dim binary token-ID substrate

In the minimalist substrate setting, each token with integer ID $t$ is mapped to a fixed binary vector $e(t) \in \{0,1\}^{16}$ given by the 16 least significant bits of $t$. To match the Transformer width, we expand $e(t)$ to $d_{\text{model}}$ by repetition (i.e., tiling the 16-dim vector to length $d_{\text{model}}$), producing a deterministic non-semantic input representation. This embedding is fully frozen throughout training.

Crucially, we operate under an approximately **constant trainable-parameter budget**, which directly constrains the optimizer-state memory footprint. At selected stages, we reallocate this budget from training a newly added layer to training LoRA adapters across the existing stack, enabling a limited form of global adjustment without unfreezing the full model. This LoRA-based fine-tuning allows the entire network to adapt to its new depth, promoting global integration of knowledge and preventing the ossification of lower layers, all without increasing the instantaneous memory requirements for training.

## 3. EXPERIMENTAL SETUP

### 3.1. Overview of Experiments

To validate our constructive learning framework, we conduct two distinct sets of experiments. The first demonstrates the **feasibility** of our method for training large-scale models under typical hardware memory constraints. The second provides a **rigorous controlled comparison** of our approach against key baselines to evaluate its performance and isolate the contribution of frozen embeddings. We distinguish two objectives: (i) minimizing peak memory via a fixed trainable-parameter budget, and (ii) evaluating whether this constraint yields competitive downstream performance at a fixed final architecture. All controlled comparisons use identical data, tokenizer, and hyperparameters unless explicitly stated.

### 3.2. Model Architectures and Baselines

All models in our experiments are standard decoder-only Transformers, similar to GPT-2, using rotary embeddings and GELU activations.

#### 3.2.1. LARGE-SCALE FEASIBILITY STUDY

For the feasibility study, we use a large model configuration with a hidden dimension $d_{\text{model}} = 4096$, $n_{\text{head}} = 32$, and a vocabulary size of 131,072. The model, which we refer to as `ABS-BVV`, is grown progressively from 1 to 6 layers. At each stage, only the newly added layer(s) are trained, keeping the number of trainable parameters below approximately 740M. This ensures the optimizer states fit within the memory of a single 80GB H100 GPU. The final 6-layer model reaches 2.3B parameters.

#### 3.2.2. CONTROLLED COMPARATIVE STUDY

For the controlled study, we use a smaller, fixed architecture of 9 layers with $d_{\text{model}} = 1024$, $n_{\text{head}} = 32$, and a vocabulary size of 65,536, resulting in models of approximately 247.6M parameters. We compare our method against three baselines and two ablations (six models total):

- **Constructive Frozen ('Model UNICODE 1_9'):** Our proposed method. It is grown incrementally in three stages (layers 1-3, then 4-6, then 7-9). The visual Unicode embedding layer is frozen from the start, as are previously trained layers at each new stage. The final model has 247.6M total parameters, of which 142.7M are frozen.

- **Monolithic Trainable ('Model unfrozen (baseline)'):** The primary baseline. A standard 9-layer Transformer with a randomly initialized, fully trainable embedding layer. It is trained end-to-end in a traditional, monolithic fashion. All 247.6M parameters are trainable.

- **Monolithic Frozen ('Model frozen UNICODE (baseline)'):** The secondary baseline. A standard 9-layer Transformer with fully frozen and Unicode-predefined embedding layer. It is trained end-to-end in a traditional, monolithic fashion. 67.1M (of 247.6M) parameters are frozen.

- **Constructive Trainable (Ablation 1, 'Model unfrozen 1_9'):** This model isolates the effect of the growth strategy itself. It is grown identically to our proposed method (in three stages), but starts with a standard, trainable embedding layer. Embeddings are trained only during the first growth stage (layers 1–3) and then frozen to preserve the input interface for frozen lower blocks. In constructive growth, previously trained blocks are frozen. Therefore, leaving token embeddings trainable in later stages would induce a distribution shift at the input of frozen blocks (layers 1–3), which cannot be compensated by those blocks. This tests whether constructive growth is effective without the frozen substrate.

- **Constructive Frozen (16-dim Binary, Ablation 2, 'Model 16_bit 1_9'):** This model tests the limits of the emergent semantics principle. It uses a minimalist, 16-dimensional binary vector for each token (derived from the token's integer ID) as its frozen embedding. This vector is projected to $d_{model} = 1024$ via simple repetition. The model is then grown constructively. The final model has 181.6M total parameters, of which 76.6M are frozen.

- **Monolithic Frozen ('Model frozen 16-bit (baseline)'):** The third baseline. A standard 9-layer Transformer with fully frozen 16-dimensional binary embedding layer. It is trained end-to-end in a traditional, monolithic fashion. 1.0M (of 181.6) parameters are frozen.

### 3.2.3. EXTENDED WEB-SCALE 16-DIM SCALING RUN

To test whether constructive growth with extremely low-capacity frozen token identifiers remains viable beyond our 4B-token controlled study, we run an extended experiment on a web-scale mixture (FineWeb-Edu + Cosmopedia) for 68.9B tokens. We grow the model in 4-layer increments ($4{\to}8{\to}12{\to}16$), freezing all previously trained blocks, while keeping the number of trainable parameters approximately constant at 117.5M (newest 4 blocks + LM head). Between growth stages, we interleave global LoRA phases applied to all layers under the same trainable-parameter budget. Full schedules and hyperparameters are reported in Appendix D.

### 3.3. Training Datasets and Protocol

**Controlled study data (4B tokens).** The controlled experiments use a 4B-token mixture (English Wikipedia subset + SFT datasets).

**Web-scale data (68.9B tokens).** The extended scaling run uses FineWeb-Edu + Cosmopedia for a total of 68.9B tokens, with interleaved instruction/SFT samples (stage-wise mixing reported in Appendix D).

## 4. EXPERIMENTS AND RESULTS

### 4.1. Results: Progressive Layer-Wise Growth

We grew a model from 1 ('abs-bvv-1') to 6 layers ('abs-bvv-6'). Figure 1 illustrates the training process. Each sharp spike in the loss corresponds to the addition of a new, untrained layer, followed by rapid convergence, demonstrating the stability of the method.

The results, summarized in Figures 2 and 3, reveal a clear pattern. General reasoning ability on MMLU increases steadily with depth, from 18.08% with one layer to 21.63% with six. More strikingly, the ability to perform complex extractive question-answering (SQuAD) is virtually non-existent in shallow models (1.21% at $n_{layer} = 1$). A significant signal appears only at 'n_layer=3' (3.75%), and reaching its peak performance in our experiments at $n_{layer} = 6$ (5.55%). We report SQuAD exact match (EM) under our evaluation setup.

We use the growth schedule as a controlled lens to track how capability indicators change as depth is increased under a fixed trainable-parameter budget.

### 4.2. Results: Controlled Comparative Study

The controlled comparative study was designed to rigorously evaluate our constructive learning method against traditional monolithic training and key ablations. The results reveal a clear trade-off between optimization on the training objective (loss) and performance on downstream reasoning tasks (MMLU).

As shown in Figure 4, all models exhibit stable convergence. The monolithic baseline, with all its parameters being trainable, unsurprisingly achieves the lowest final training loss. This is expected because monolithic training updates all parameters directly; our goal is instead to evaluate downstream performance under constrained trainability. In contrast, the constructive models, particularly those with frozen components, plateau at a slightly higher loss. This is an expected outcome for two key reasons: first, a significant

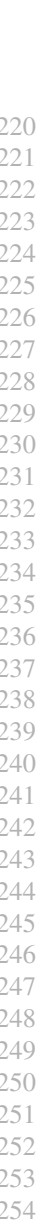
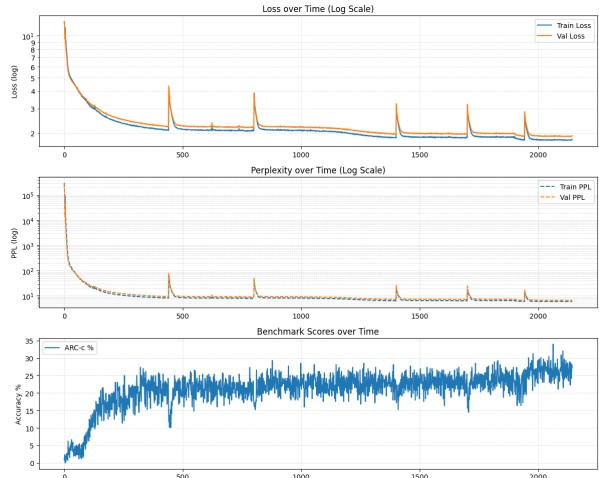

*Figure 1.* Training dynamics during progressive layer-wise growth. Each loss spike marks the stacking of a new layer, followed by rapid convergence. We report loss spikes at layer addition and subsequent recovery; we also track ARC-c as an auxiliary indicator, which tends to improve with depth in this run.

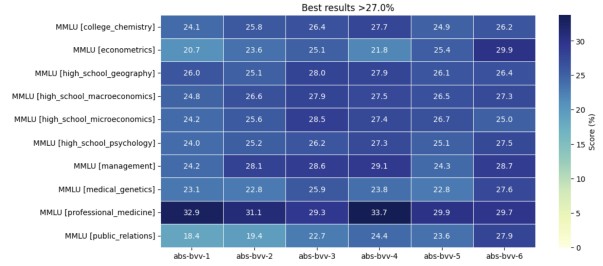

*Figure 2.* Benchmark performance as a function of model depth. Note the significant jump in SQuAD score at 'n_layer=3', indicating the onset of measurable extractive QA performance in this setup.

*Figure 3.* MMLU performance on select subjects as a function of model depth, illustrating how different reasoning capabilities strengthen as the model grows.

portion of their parameters are not directly optimized for the training objective; second, and critically, this controlled study intentionally omits the LoRA-based holistic tuning stages used in our large-scale experiment. This was done to strictly isolate the raw effect of the layer-stacking methodology.

The key finding of this study is presented in Figure 5. On MMLU, 'Model UNICODE 1_9' is competitive with the monolithic baseline in this setting. Importantly, the effect is task-dependent: on ARC and commonsense-oriented metrics, differences are smaller and often within a narrow band. We therefore interpret MMLU gains as an empirical signal that layer-wise growth on a frozen substrate can be competitive for some reasoning-heavy evaluations, rather than as a universal improvement claim.

Results are from single runs due to compute constraints; we focus on stability patterns and cross-scale replication (248M and 2.3B) rather than statistical variance.

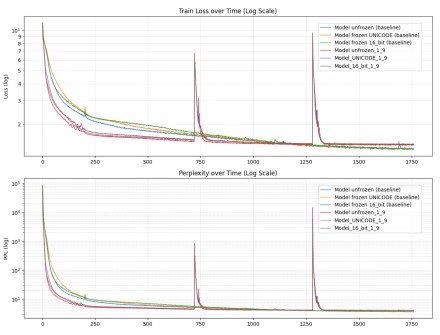

*Figure 4.* Training dynamics during progressive layer-wise growth for 'Model unfrozen (baseline)', 'Model frozen UNICODE (baseline)', 'Model frozen 16_bit (baseline)', 'Model UNICODE 1_9', 'Model unfrozen 1_9' and 'Model 16_bit 1_9'. Each loss spike marks the stacking of a new layer group, followed by rapid convergence.

The ablation studies provide further crucial insights. The 'Model unfrozen 1_9' underperforms the monolithic baseline and the frozen-substrate growth model on MMLU in this setup, suggesting that the growth schedule alone is not sufficient to recover the same behavior. This points to an interaction between the growth procedure and the presence of a stable embedding substrate. Remarkably, the 'Model 16_bit 1_9', despite using extremely impoverished 16-dim binary embeddings, still achieves competitive performance, underscoring the Transformer's powerful ability to induce semantic structure from even minimal, consistent input signals.

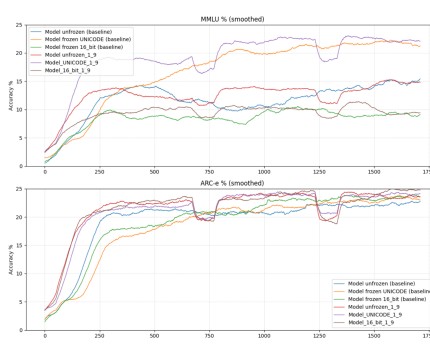

*Figure 5.* MMLU and ARC-E metric dynamics during progressive layer-wise growth for 'Model unfrozen (baseline)', 'Model frozen UNICODE (baseline)', 'Model frozen 16_bit (baseline)', 'Model UNICODE 1_9', 'Model unfrozen 1_9' and 'Model 16_bit 1_9'.

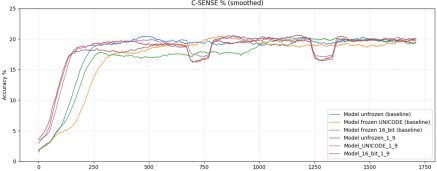

*Figure 6.* C-SENSE metric dynamics during progressive layer-wise growth for 'Model unfrozen (baseline)', 'Model frozen UNICODE (baseline)', 'Model frozen 16_bit (baseline)', 'Model UNICODE 1_9', 'Model unfrozen 1_9' and 'Model 16_bit 1_9'.

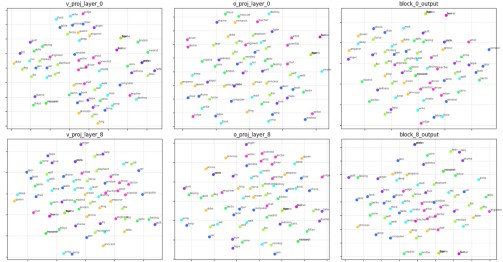

*Figure 7.* t-SNE visualization for the first and the last layer attention projections (v_proj and o_proj), and transformer block outputs for 'Model UNICODE 1_9' with frozen embedding layer. In this picture semantic groups (e.g., numbers, professions, animals) tend to make clusters **after** frozen embedding layer demonstrating emergence of semantic structure.

These performance differences are mirrored in the models' internal representations. Visualizations of the final layers (Figure 7) show that rich semantic clusters (e.g., numbers, professions) clearly emerge within the deeper layers of 'Model UNICODE 1_9'. These visualizations are consistent with the hypothesis that semantic organization can arise in deeper layers even when the input embeddings are non-semantic; however, we treat t-SNE as a qualitative illustration rather than definitive proof.

Due to compute limits we report single runs; however, the qualitative growth dynamics replicate across two scales (248M and 2.3B).

### 4.3. Extended 16-dim scaling run (16 layers, 269.7M params, 68.9B training tokens).

To test whether constructive growth remains viable with extremely low-capacity frozen token identifiers beyond our 4B-token controlled study, we ran an additional experiment on a larger web-scale mixture (FineWeb-Edu + Cosmopedia) for 68.9B tokens. We grow the model in *4-layer* increments (4→8→12→16), freezing all previous blocks, while keeping the number of trainable parameters approximately constant at 117.5M (newest 4 blocks + LM head). We also interleave global LoRA adaptation phases under the same trainable-parameter budget. This run reaches 28.92% MMLU and 19.10% SQuAD EM with a 16-layer (269.7M base) model using frozen 16-dimensional binary token-ID embeddings. Full details, hyperparameters, and stage-wise metrics are reported in Appendix D.

In summary, this controlled study demonstrates that constructive growth on a frozen substrate is not only a viable training strategy and can be competitive with monolithic training on reasoning-heavy evaluations in this setup, by allowing the model to more effectively specialize its deeper layers for semantic composition.

*Table 1.* Extended 16-dim scaling run (FineWeb-Edu + Cosmopedia, 4×H100). Trainable parameters are kept fixed at ≈117.5M across stages (newest 4 blocks + LM head, or LoRA adapters).

| Stage | Tokens (B) | MMLU (%) | SQuAD EM (%) |
|---|---|---|---|
| 1–4 Growth | 15.5 | 16.95 | 1.21 |
| 1–8 Growth | 31.0 | 19.42 | 2.77 |
| 1–8 LoRA | 38.8 | 21.31 | 5.39 |
| 1–12 Growth | 54.3 | 23.38 | 9.45 |
| 1–12 LoRA | 59.2 | 23.90 | 12.19 |
| 1–16 Growth | 65.5 | 24.58 | 13.48 |
| 1–16 LoRA | 68.9 | 28.92 | 19.10 |

## 5. DISCUSSION

Our findings present a compelling case for a constructive approach to building LLMs, moving away from the monolithic paradigm.

### 5.1. The Frozen Embedding as a Universal Docking Port

The key enabler for our methods is the shared, frozen representational substrate. A shared frozen embedding layer can act as a standardized interface for controlled comparisons across training procedures and for incremental construction, because it reduces one source of drift (input representation) when components are frozen or composed.

### 5.2. From Monolithic Training to Constructive Growth

Layer-wise growth provides a structured way to increase depth while limiting the number of trainable parameters at any time. This can be useful both as an engineering technique under memory constraints and as an experimental tool to probe how capabilities change with depth.

### 5.3. Implications for a Modular AI Ecosystem

This paradigm has significant implications for the future of AI development:

- Resource Efficiency and Specialization: Organizations could train smaller, expert models on proprietary or specialized data.

- Continual Learning: New knowledge and skills may be added by training new modules or extending depth with reduced interference compared to full-model updates, though a full continual-learning evaluation is outside the scope of this work.

- **Synergy with Model Quantization:** Our approach, particularly with the minimalist 16-dimensional binary embeddings, introduces a "quantization-native" input layer. Standard models learn continuous, high-precision embeddings that must be aggressively quantized post-training. In contrast, our method is built upon an inherently discrete foundation: the embedding vectors themselves are derived from binary, monochrome glyph images. The discrete nature of the embedding substrate is compatible with low-precision representations and invites future research on whether such substrates ease quantization-aware training or reduce sensitivity to aggressive post-training quantization.

- Democratization: This approach lowers the barrier to entry. Instead of requiring the resources to train a 100 B+ parameter model from scratch, researchers could contribute to a larger ecosystem by developing and sharing smaller, compatible modules.

### 5.4. Limitations

Our controlled comparisons are based on single runs due to compute limits; we therefore emphasize stability patterns and cross-scale replication rather than variance estimates. The extended web-scale run includes stage-wise instruction/SFT mixing, which may confound purely unsupervised pretraining comparisons.

## 6. CONCLUSION

We presented a constructive layer-wise growth methodology for decoder-only Transformers built on a frozen embedding substrate. Across a feasibility study and a controlled comparison, we show stable optimization dynamics under a fixed trainable-parameter budget and competitive downstream performance in selected evaluations. These results motivate further work on scaling, robustness across tasks, and quantitative analyses of representation emergence.

## 7. IMPACT STATEMENT

This paper studies training procedures for decoder-only Transformer language models under memory constraints by freezing most parameters and training newly added layers (and optionally lightweight adapters). **Potential benefits** include reducing peak optimizer-state memory, enabling experimentation for smaller labs, and lowering energy/compute requirements for incremental scaling. **Potential risks** are similar to those of language model scaling in general: making training more accessible may also lower the barrier for creating models that could be misused (e.g., for misinformation or spam). We do not introduce new personally-identifying datasets, and our work focuses on training methodology rather than deployment.

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

## A. APPENDIX A: Training loss and perplexity data

*Table 2.* Training loss and perplexity data

| Step | Model unfrozen (base line) Loss | Model unfrozen (base line) PPL | Model frozen UNI CODE (base line) Loss | Model frozen UNI CODE (base line) PPL | Model frozen 16_bit (base line) Loss | Model frozen 16_bit (base line) PPL | Model unfrozen 1_9 Loss | Model unfrozen 1_9 PPL | Model UNI CODE 1_9 Loss | Model UNI CODE 1_9 PPL | Model 16_bit 1_9 Loss | Model 16_bit 1_9 PPL |
|---|---|---|---|---|---|---|---|---|---|---|---|---|
| 0 | 11.34 | 84005 | 11.12 | 67363 | 11.40 | 89155 | 10.88 | 53254 | 10.67 | 43094 | 11.32 | 82104 |
| 050000 | 2.90 | 18.27 | 3.35 | 28.62 | 3.11 | 22.50 | 2.21 | 9.11 | 2.36 | 10.63 | 2.37 | 10.73 |
| 100000 | 2.29 | 9.82 | 2.47 | 11.85 | 2.38 | 10.85 | 1.72 | 5.61 | 1.76 | 5.79 | 1.84 | 6.29 |
| 150000 | 2.00 | 7.38 | 2.14 | 8.48 | 2.15 | 8.60 | 1.63 | 5.08 | 1.65 | 5.22 | 1.69 | 5.43 |
| 200000 | 1.86 | 6.45 | 2.00 | 7.35 | 2.06 | 7.85 | 1.59 | 4.93 | 1.61 | 5.03 | 1.66 | 5.24 |
| 250000 | 1.77 | 5.90 | 1.85 | 6.37 | 1.96 | 7.10 | 1.56 | 4.78 | 1.60 | 4.94 | 1.60 | 4.95 |
| 300000 | 1.71 | 5.52 | 1.75 | 5.76 | 1.84 | 6.28 | 1.53 | 4.62 | 1.55 | 4.73 | 1.57 | 4.83 |
| 350000 | 1.66 | 5.24 | 1.70 | 5.45 | 1.74 | 5.71 | 1.50 | 4.48 | 1.53 | 4.63 | 1.54 | 4.67 |
| 400000 | 1.60 | 4.98 | 1.63 | 5.11 | 1.66 | 5.26 | 1.47 | 4.37 | 1.51 | 4.51 | 1.52 | 4.58 |
| 450000 | 1.54 | 4.69 | 1.56 | 4.77 | 1.58 | 4.86 | 1.45 | 4.27 | 1.49 | 4.41 | 1.49 | 4.44 |
| 500000 | 1.50 | 4.48 | 1.50 | 4.49 | 1.53 | 4.63 | 1.44 | 4.22 | 1.47 | 4.33 | 1.46 | 4.32 |
| 550000 | 1.46 | 4.29 | 1.47 | 4.34 | 1.49 | 4.42 | 1.42 | 4.12 | 1.44 | 4.22 | 1.44 | 4.22 |
| 600000 | 1.40 | 4.06 | 1.40 | 4.04 | 1.43 | 4.17 | 1.42 | 4.12 | 1.44 | 4.21 | 1.43 | 4.19 |
| 650000 | 1.38 | 3.96 | 1.39 | 4.02 | 1.39 | 4.03 | 2.25 | 9.45 | 2.02 | 7.53 | 2.02 | 7.54 |
| 700000 | 1.35 | 3.87 | 1.37 | 3.92 | 1.37 | 3.93 | 1.41 | 4.09 | 1.43 | 4.18 | 1.42 | 4.14 |
| 750000 | 1.33 | 3.78 | 1.34 | 3.80 | 1.35 | 3.85 | 1.41 | 4.09 | 1.43 | 4.17 | 1.42 | 4.15 |
| 800000 | 1.31 | 3.71 | 1.32 | 3.75 | 1.35 | 3.86 | 1.40 | 4.06 | 1.42 | 4.14 | 1.42 | 4.16 |
| 850000 | 1.33 | 3.77 | 1.31 | 3.71 | 1.31 | 3.72 | 1.40 | 4.04 | 1.42 | 4.12 | 1.43 | 4.16 |

## B. APPENDIX B: Parameter Details for Large-Scale Model Growth

Table 3 provides a detailed breakdown of the parameter counts for the large-scale model growth experiment. The base architecture is $d_{\mathrm{model}} = 4096$, $n_{\mathrm{head}} = 32$, and a vocabulary size of 131,072.

A key aspect of our methodology is maintaining a constant budget of trainable parameters at each stage, constrained to approximately 740M to fit within the memory of a single H100 accelerator. This budget is allocated in two strategic ways:

- **Layer Growth:** In the primary growth stages, the budget is dedicated entirely to training the parameters of a newly added top layer, while all preceding layers remain frozen.

- **Holistic LoRA Tuning:** At later stages (e.g., after 3 and 5 layers), the same budget is reallocated to train low-rank adapters applied to *all* layers simultaneously. The LoRA rank and target modules were chosen to ensure the total number of trainable adapter parameters also fit within the ≈740M budget. This allows for a global, computationally efficient fine-tuning of the entire model stack to better integrate the new layers without increasing the instantaneous memory footprint.

*Table 3.* Layer-wise growth parameters for the large-scale (ABS) model. Each training stage operates under a fixed budget of $\approx$740M trainable parameters. This budget is allocated either to a new top layer (Method: Growth) or to LoRA adapters across the entire stack (Method: Fine-tuning). Total Params incl. Adapters(M) counts temporary LoRA adapter parameters present during the LoRA stages; the underlying base model size at ABS-6 is 2.3B parameters.

| Model Stage | Layers | Method | Total Params incl. Adapters(M) | Trainable (M) | Frozen (M) |
|---|---|---|---|---|---|
| ABS-1 | 1 | Growth | 1275.2 | 738.4 | 536.9 |
| ABS-2 | 2 | Growth | 1476.6 | 738.4 | 738.2 |
| ABS-3 | 3 | Growth | 1677.9 | 738.4 | 939.5 |
| ABS-3 (LoRA) | 3 | Fine-tuning | 1728.3 | 738.4 | 989.9 |
| ABS-4 | 4 | Growth | 1879.3 | 738.4 | 1140.9 |
| ABS-5 | 5 | Growth | 2080.7 | 738.4 | 1342.3 |
| ABS-5 (LoRA) | 5 | Fine-tuning | 2812.0 | 738.4 | 2073.6 |
| ABS-6 | 6 | Growth | 2282.0 | 738.4 | 1543.6 |
| ABS-6 (LoRA) | 6 | Fine-tuning | 3008.7 | 726.6 | 2270.3 |

## C. APPENDIX C: Parameter Details for Controlled Study Models

Table 4 provides the corresponding parameter breakdown for the smaller-scale models used in our controlled comparative study. The base architecture for these models is $d_{\mathrm{model}} = 1024$, $n_{\mathrm{head}} = 32$, and a vocabulary size of 65,536. The models were grown in stages of three layers each, with previously trained layers being frozen at subsequent stages.

*Table 4.* Parameter breakdown for the controlled study models (final size $\approx$248M).

| Model Type | Stage (Layers) | Total (M) | Trainable (M) | Frozen (M) |
|---|---|---|---|---|
| Constructive (UNICODE) *(Our Method)* | 1-3 | 172.1 | 105.0 | 67.1 |
| | 1-6 | 209.8 | 105.0 | 104.9 |
| | **1-9 (Final)** | **247.6** | **105.0** | **142.7** |
| Constructive (Trainable Emb.) *(Constructive trainable)* | 1-3 | 172.1 | 172.1 | 0.0 |
| | 1-6 | 209.8 | 105.0 | 104.9 |
| | **1-9 (Final)** | **247.6** | **105.0** | **142.7** |
| Constructive (16-dim Emb.) *(Ablation)* | 1-3 | 106.0 | 105.0 | 1.0 |
| | 1-6 | 143.8 | 105.0 | 38.8 |
| | **1-9 (Final)** | **181.6** | **105.0** | **76.6** |
| Monolithic (Trainable Emb.) *(Baseline trainable classic)* | **1-9 (Final)** | **247.6** | **247.6** | **0.0** |
| Monolithic (Frozen UNICODE Emb.) *(Baseline)* | **1-9 (Final)** | **247.6** | **180.5** | **67.1** |
| Monolithic (Frozen 16_bit Emb.) *(Baseline)* | **1-9 (Final)** | **181.6** | **180.5** | **1.0** |

## D. APPENDIX D: Extended 16-dim (n_embed = 16) scaling run on FineWeb-Edu + Cosmopedia

**Setup.** We train a decoder-only Transformer with $d_{\mathrm{model}} = 1024$, $n_{\mathrm{head}} = 32$, dropout=0.05, and a frozen 16-dimensional binary token-ID embedding (expanded to $d_{\mathrm{model}}$ by repetition). Training proceeds via constructive growth in 4-layer increments. At each growth stage, previously trained blocks remain frozen; only the newest 4 blocks and the LM head are trained, keeping trainable parameters fixed at $\approx$117.5M. Between growth stages we optionally run a LoRA phase applied to all blocks, choosing ranks such that the total number of trainable LoRA parameters also stays within $\approx$117.5M. The training mixture is based on FineWeb-Edu and Cosmopedia (web-scale), with an interleaved instruction-tuning (SFT) dataset

sampled either periodically or with probability $p_{\text{sft}}$ (see Table 7).

*Table 5.* Extended 16-dim (n_embed = 16) scaling run: stage schedule, parameter counts, and training budget. Trainable parameters are kept approximately constant at $\approx$117.5M across stages (either newest 4 blocks + LM head, or LoRA adapters). Tokens seen are taken from the training log (end of stage).

| Stage | Layers | Method | Total (M) | Trainable (M) | Frozen (M) | Tokens seen (B) |
|-------|--------|--------|-----------|---------------|------------|-----------------|
| 1–4   | 4      | Growth | 118.6     | 117.5         | 1.0        | 15.5            |
| 1–8   | 8      | Growth | 169.0     | 117.5         | 51.4       | 31.0            |
| 1–8   | 8      | LoRA   | 286.5     | 117.5         | 169.0      | 38.8            |
| 1–12  | 12     | Growth | 219.3     | 117.5         | 101.8      | 54.3            |
| 1–12  | 12     | LoRA   | 336.8     | 117.4         | 219.3      | 59.2            |
| 1–16  | 16     | Growth | 269.7     | 117.5         | 152.2      | 65.5            |
| 1–16  | 16     | LoRA   | 387.1     | 117.4         | 269.7      | 68.9            |

*Table 6.* Extended 16-dim scaling run: downstream metrics by stage (reported as mean $\pm$ uncertainty as logged during training).

| Stage | MMLU | SQuAD |
|-------|------|-------|
| 1–4 Growth  | $16.95 \pm 0.16$ | $1.21 \pm 0.40$  |
| 1–8 Growth  | $19.42 \pm 0.16$ | $2.77 \pm 0.48$  |
| 1–8 LoRA    | $21.31 \pm 0.14$ | $5.39 \pm 0.62$  |
| 1–12 Growth | $23.38 \pm 0.16$ | $9.45 \pm 0.83$  |
| 1–12 LoRA   | $23.90 \pm 0.19$ | $12.19 \pm 0.80$ |
| 1–16 Growth | $24.58 \pm 0.18$ | $13.48 \pm 0.94$ |
| 1–16 LoRA   | $28.92 \pm 0.19$ | $19.10 \pm 0.95$ |

*Table 7.* Key hyperparameters for the extended 16-dim scaling run (stage-wise). We keep the effective token batch approximately constant by adjusting gradient accumulation when micro-batch size changes.

| Stage | Micro-batch | Grad accum | LR | SFT mixing |
|-------|-------------|------------|-----|------------|
| 1–4 Growth  | 108 | 32 | $1.05 \cdot 8e{-}4$   | periodic (every 250 iters) |
| 1–8 Growth  | 108 | 32 | $0.95 \cdot 8e{-}4$   | periodic (every 250 iters) |
| 1–8 LoRA    | 64  | 54 | $0.50 \cdot 8e{-}4$   | periodic (every 250 iters) |
| 1–12 Growth | 108 | 32 | $0.45 \cdot 8e{-}4$   | $p_{\text{sft}} = 0.004$ |
| 1–12 LoRA   | 48  | 72 | $0.25 \cdot 8e{-}4$   | $p_{\text{sft}} = 0.01$  |
| 1–16 Growth | 104 | 32 | $0.10 \cdot 8e{-}4$   | $p_{\text{sft}} = 0.05$  |
| 1–16 LoRA   | 36  | 96 | $0.055 \cdot 8e{-}4$  | $p_{\text{sft}} = 0.33$  |

