# Emergent Semantics Beyond Token Embeddings: Transformer LMs with Frozen Visual Unicode Representations

## Abstract

Understanding the locus of semantic representation in large language models (LLMs) is crucial for interpretability and architectural innovation. The dominant paradigm posits that trainable input embeddings serve as foundational "meaning vectors." This paper challenges that view. We construct Transformer models where the embedding layer is entirely frozen, with vectors derived not from data, but from the visual structure of Unicode glyphs. These non-semantic, precomputed visual embeddings are fixed throughout training. Our method is compatible with any tokenizer, including a novel Unicode-centric tokenizer we introduce to ensure universal text coverage. Despite the absence of trainable, semantically initialized embeddings, our models converge, generate coherent text, and, critically, outperform architecturally identical models with trainable embeddings on the MMLU reasoning benchmark. We attribute this to "representational interference" in conventional models, where the embedding layer is burdened with learning both structural and semantic features. Our results indicate that high-level semantics are not inherent to input embeddings but are an emergent property of the Transformer's compositional architecture and data scale. This reframes the role of embeddings from meaning containers to structural primitives. We release all code and models to foster further research.

## 1 INTRODUCTION

Understanding where and how semantic abstractions arise in transformer language models is both a theoretical and practical concern for the development of scalable, robust, and interpretable AI systems. Traditionally, input token embeddings—learned representations from large corpora—are viewed as the primary locus of meaning: their algebraic properties have been cited as evidence of encodable semantic relationships ("king - man + woman = queen"), and advancing architectures routinely attribute model improvements to smarter or larger embedding matrices. This paradigm has shaped assumptions about what is required for "intelligent" representation learning.

The robustness of modern Large Language Models (LLMs) to orthographic variations, such as in 'I can wRiTe', highlights a fundamental question. The model's understanding is not derived from a single, semantically rich token for "write". Instead, it often relies on composing meaning from a sequence of semantically poor character- or subword-level tokens (e.g., 'w', 'R', 'i', 'T', 'e'). If the input vectors for these atomic units lack inherent semantic content, where does the model's understanding originate? This paper argues and empirically demonstrates that semantics are not a property of input embeddings but an emergent phenomenon of the Transformer architecture itself.

However, recent advances in byte and character-level modeling, as well as modular and multi-lingual architectures, suggest that the relationship between embeddings and high-level meaning is far subtler. In pursuit of a deeper understanding, we investigate the extreme case: Transformer language models where the input embedding layer is never trained and, in fact, deliberately devoid of semantic optimization. Instead, we fix this layer to a deterministic mapping based on visual Unicode glyph representations and token-level n-gram image composites.

Our experimental framework draws embeddings not from data-driven optimization, but from precomputed character images spanning the entire Unicode range. For multicharacter tokens (as in byte-pair and unigram models), we employ compositional image techniques to generate a fixed-length visual feature vector per token, achieving universal text coverage. The tokenizer is similarly Unicode-centric, balancing character-level granularity with multi-script n-gram tokens to maximize overlap across languages.

Instead of pursuing state-of-the-art benchmark scores, which often depend on massive scale, our primary goal is to isolate and test a fundamental hypothesis: can language abstraction emerge without trainable, semantically-rich input embeddings? We deliberately constrain model and dataset sizes to create a controlled environment for comparing the learning dynamics of frozen- versus trainable-embedding models. Our focus is on the conditions under which meaning emerges, not on outperforming large-scale systems.

Critically, we observe that despite this radical departure from conventional practice, these "frozen-embedding" models not only converge but display robust convergence and, surprisingly, outperform architecturally identical counterparts with trainable embeddings on reasoning benchmarks like MMLU, suggesting representational interference in conventional approaches. This suggests that emergent abstraction appears within the transformer blocks themselves, implying that semantic structure is an architectural phenomenon, not an initialization artifact. Our main contributions are:

- demonstrate that semantic understanding in Transformers can emerge without trainable input embeddings

- introduce a universal visual embedding scheme compatible with any tokenizer

- identify "representational interference" as a potential limitation of trainable embeddings

- release open-source implementations enabling reproducibility

The rest of this paper is organized as follows: Section II reviews related work; Section III details our method for constructing visual embeddings and Unicode tokenization; Section IV describes the experimental setup; Section V presents results and analysis; Section VI discusses implications for future architectures; and Section VII concludes. Code and all artifacts are released to foster further progress in both science and industry.[1]

## 2 RELATED WORK

Semantic representation in neural language models has received extensive attention. Early word2vec (Mikolov et al., 2013) and GloVe (Pennington et al., 2014) models established the conceptual centrality of "meaningful" dense vector spaces, with well-documented algebraic properties in vector arithmetic. Subsequent transformer-based LMs (Radford et al., 2019) extended this paradigm to context-dependent embeddings, further entrenching the notion that trainable input vectors are the source of semantic capability. However, several lines of research challenge this view.

Character- and byte-level models, such as ByT5 (Xue et al., 2022) and CANINE (Clark et al., 2022), forego word-level tokenization for universal coverage; yet, even here, embedding matrices are trainable and morph into semantic vector spaces during pretraining. Cross-lingual and modular approaches pursue shared embedding spaces for transfer (Conneau et al., 2018; Lample & Conneau, 2019), but rely on substantial alignment learning. Mixture-of-experts (Shazeer et al., 2017; Fedus et al., 2022) and adapter fusion (Pfeiffer et al., 2020) aim to modularize reasoning, yet remain dependent on trainable embeddings. Our input embedding matrix is never trained and strictly visual/surface oriented.

More radically, recent work in visual-language models (Tan & Bansal, 2022) investigates multimodal representations by grounding language in visual contexts. Models like CLIP (Radford et al., 2021) or Flamingo (Alayrac et al., 2022) learn joint embeddings for text and images. However, these approaches typically combine separate, powerful encoders for each modality and aim to align two rich semantic streams, rather than enforcing a non-semantic, frozen vector space for language as we propose.

---

[1]Our code and models are provided in the supplementary materials.

The idea of incorporating visual information from text glyphs is not entirely new. Prior work has successfully utilized visual features to enhance specific tasks. For instance, visual representations have been used to improve machine translation, especially for handling rare words and achieving robust cross-lingual transfer Salesky et al. (2023); Wang et al. (2020). Other studies have leveraged glyph-aware embeddings for better representation of logographic languages like Chinese Dai & Cai (2017) or to learn character-level compositionality Liu et al. (2017).

Our work differs from these approaches in two fundamental aspects. First, whereas prior studies use visual features as a trainable component or an auxiliary signal to enrich semantic embeddings, our methodology employs them as a completely frozen, non-semantic substrate. Second, our primary goal is not to improve a specific downstream task, but to use this constraint as a tool to investigate a foundational question: whether semantic understanding can emerge exclusively within the Transformer's compositional layers, without any semantic information provided at the input. In our work, this reframes the role of visual features from an enhancement to a basic, structural primitive.

## 3 METHOD

### 3.1 FROZEN VISUAL UNICODE EMBEDDINGS

Our method generates a fixed embedding matrix E of shape (V, d_model), where V is the vocabulary size and d_model is the model's hidden dimension. The vector for each token is derived through a deterministic, multi-step process:

**Glyph Rendering:** Each token in the vocabulary is rendered into a bitmap. Single-character tokens are rendered directly. Multi-character n-gram tokens are rendered by horizontally concatenating the glyphs of their constituent characters. For example, the token "ing" is rendered as a single image containing the sequence of 'i', 'n', 'g' glyphs. We use a standardized font (e.g., unifont-14.0.01) at a fixed-point size.

**Image Standardization:** The resulting variable-width bitmaps are resized to a fixed square resolution (e.g., 32x32 for d_model=1024) using bilinear interpolation. This creates a standardized visual representation for every token.

**Vectorization:** Each standardized HxH bitmap is flattened into a $H^2$-dimensional raw vector (binary components).

**Projection:** To project the high-dimensional image vector into the model's latent space, we apply Principal Component Analysis (PCA). A PCA model is fit on the raw vectors of the entire vocabulary, and the top d_model principal components are used to transform each raw vector into a d_model-dimensional embedding. This preserves the maximum variance of the visual features within the target dimensionality (floating points components). PCA was chosen for its determinism, computational efficiency, and ability to capture dominant visual variance in a fixed-dimensional space without introducing trainable parameters. This aligns with our core hypothesis of investigating emergent semantics absent of any data-driven feature learning at the input layer. While a learned visual encoder like a CNN could potentially extract richer features, it would re-introduce trainable parameters, confounding our experimental goal of isolating the Transformer's compositional layers.

**Normalization and Freezing:** The resulting embedding vectors are L2-normalized. The final matrix E is loaded into the Transformer's embedding layer, and its weights are frozen (i.e., requires_grad=False) for all subsequent training stages.

The approach supports tokenizers from arbitrary LMs (e.g., Mistral Nemo tokens text used in bvv241-nemo tokenizer), as new tokens are mapped via their textual form.

To investigate the influence of PCA and normalization, as well as to understand the fundamental significance of visual representations of token texts, a maximally simplified approach was implemented. For a vocabulary of 65,536 tokens, a frozen pre-computed tensor was constructed based on binary components of a 16-dimensional vector for each token—essentially, each token was represented by its index as a 16-dimensional vector where each component is a single bit, thereby eliminating the influence of PCA algorithms, normaliza-

tion, and additional information about the visual representation of the token text (model 'Model_16_bit'). Similarly, to study the influence of PCA and normalization algorithms separately, another pre-computed tensor of 16-dimensional vector representations was constructed, but this time with the application of PCA and normalization (model 'Model_16_float')

For ablation study purposes, three additional variants with binary components based on noise were created for 64-dimensional, 256-dimensional, and 1024-dimensional vectors. Three corresponding variants with applied PCA and normalization were also developed to investigate the impact on convergence and metrics in the absence of information about the visual representation of the token text itself. ('Model_1024_bit', 'Model_1024_float', 'Model_256_bit', 'Model_256_float', 'Model_64_bit', 'Model_64_float').

## 3.2 UNICODE-CENTRIC TOKENIZER CONSTRUCTION

To test our hypothesis across diverse languages and existing token schemes, we required a method to generate visual embeddings for any given vocabulary. We developed a Unicode-centric tokenization framework, bvv241, to serve this purpose.

### 3.2.1 N-gram Extraction

The first part is n-grams extraction: automatic extraction of the most frequent n-grams (sequences of n characters) from Wikipedia and Wiki-source text corpora across 32 world languages. For each language, the algorithm extracts n-grams of four different sizes (n = 2, 3, 4, 5 characters), counts their frequency using the Counter data structure, and preserves the top 1000 most frequent n-grams for each value of n. Determination of whether a character belongs to CJK languages is performed by checking if it falls within Unicode ranges: U+4E00–U+9FFF for Chinese ideographs, U+3040–U+309F for Hiragana, U+30A0–U+30FF for Katakana, and U+AC00–U+D7AF for Korean Hangul. Results are saved in a structured format containing the language code, its full name, and a dictionary with top n-grams for each length, enabling subsequent use of this data for language identification tasks, morphological pattern analysis, or language model creation.

### 3.2.2 Tokenizer Implementation and Vocabulary

The second part is tokenizer implementation: algorithm implements the creation of a multilingual tokenizer with a fixed vocabulary size of 65,536 tokens, based on statistically significant n-grams from the previous stage. The algorithm employs a direct mapping strategy for Unicode characters in the range 0x0000–0xD7FF (55,296 positions) to corresponding token indices, ensuring complete coverage of the Unicode Basic Multilingual Plane. The most frequent n-grams (8,177 unique sequences in total) are distributed across two ranges: the first 2,048 n-grams are placed in the Unicode surrogate pairs zone (0xD800–0xDFFF), which is not used for regular characters, while the remaining ones occupy the range 0xE100–0xF8F5. Special tokens, including sequence start and end markers, dialogue markup tokens, and technical markers, are reserved in the range 0xE000–0xE0FF (256 positions). Unused positions are filled with dummy tokens to achieve the target vocabulary size of 65,536 elements.

So, the core approach is based on bijective mapping of Unicode codepoints to individual tokens, whenever possible, ensuring a 1:1 relationship between characters and tokens across a wide range of scripts. Reserved, surrogate, and private Unicode ranges are systematically exploited to store custom and multi-character tokens, allowing for efficient representation of commonly occurring substrings and entire words—particularly beneficial for high-frequency terms shared across models and languages.

### 3.2.3 Vocabulary Expansion

The next version or expansion of the tokenizer from 65,536 to 131,072 tokens represents a qualitative transition from basic Unicode coverage to a hybrid architecture capable of efficiently processing modern multilingual corpora and specialized domains. In the base 65,536-token version, virtually the entire vocabulary is occupied by direct Unicode character mapping (approximately 55,000 positions) and statistically identified n-grams (about 8,000), leaving minimal space for additional optimizations. Doubling the vocabulary size to 131,072 preserves all advantages of bijective Unicode mapping while simultaneously integrating approx-

imately 19,000 high-frequency multi-character tokens extracted from analysis of leading language models. These additional tokens include whole words, stable collocations, technical terms, and programming constructs that frequently appear across various languages and domains. This approach dramatically reduces tokenization fragmentation for specialized texts, code, and low-resource languages, where traditional BPE tokenizers may split semantically coherent units into numerous small fragments.

Architecturally, the extended tokenizer utilizes the additional address space 0x10000–0x1FFFF to accommodate these high-frequency patterns, systematically exploiting reserved and private Unicode ranges. This achieves an optimal balance between universality (every Unicode character is guaranteed to have a token) and efficiency (frequent sequences are encoded with a single token). The hybrid design is particularly effective for processing mixed texts containing code-switching, technical documentation with embedded code snippets, and texts in non-Latin scripts, where standard tokenizers often demonstrate suboptimal performance due to bias toward English-language corpora during training.

| | DeepSeek-R1 | GPT-2 | GPT-4 | Mistral | Qwen | bvv241-2-3 | bvv241-max | bvv241-nemo |
|---|---|---|---|---|---|---|---|---|
| Arabic | 2.46 | 1.03 | 1.43 | 2.83 | 2.37 | 1.44 | 1.72 | 2.83 |
| Armenian | 1.82 | 0.59 | 0.59 | 2.48 | 1.05 | 1.49 | 1.79 | 2.48 |
| Azerbaijani | 2.17 | 1.60 | 2.05 | 2.66 | 2.09 | 1.55 | 1.94 | 2.66 |
| Bangla | 2.24 | 0.51 | 0.83 | 2.28 | 0.96 | 1.47 | 1.81 | 2.28 |
| Chinese | 1.51 | 0.56 | 0.86 | 1.09 | 1.36 | 1.11 | 1.20 | 1.09 |
| Czech | 2.66 | 1.91 | 2.26 | 2.74 | 2.21 | 1.64 | 1.94 | 2.74 |
| Danish | 3.05 | 2.61 | 2.98 | 3.06 | 2.87 | 1.72 | 2.11 | 3.06 |
| Dutch | 3.37 | 2.63 | 3.11 | 3.32 | 3.01 | 1.79 | 2.19 | 3.32 |
| English | 4.21 | 4.15 | 4.07 | 3.96 | 3.95 | 1.74 | 2.27 | 3.96 |
| Finnish | 2.78 | 2.32 | 2.58 | 2.86 | 2.49 | 1.80 | 2.21 | 2.86 |
| French | 3.45 | 2.79 | 3.30 | 3.65 | 3.22 | 1.72 | 2.10 | 3.65 |
| Georgian | 1.43 | 0.43 | 0.61 | 2.07 | 1.08 | 1.53 | 1.82 | 2.07 |
| German | 3.40 | 2.66 | 3.23 | 3.59 | 3.10 | 1.77 | 2.19 | 3.59 |
| Greek | 2.15 | 0.95 | 1.17 | 2.47 | 1.20 | 1.23 | 1.54 | 2.47 |
| Hebrew | 2.06 | 0.90 | 1.06 | 2.15 | 2.33 | 1.31 | 1.57 | 2.15 |
| Hindi | 1.76 | 0.68 | 1.04 | 2.41 | 1.11 | 1.30 | 1.56 | 2.41 |
| Hungarian | 2.63 | 1.95 | 2.32 | 2.88 | 2.27 | 1.71 | 2.09 | 2.88 |
| Indonesian | 3.65 | 2.69 | 3.26 | 3.55 | 3.16 | 1.75 | 2.24 | 3.55 |
| Italian | 3.40 | 2.75 | 3.23 | 3.47 | 3.11 | 1.74 | 2.14 | 3.47 |
| Japanese | 1.43 | 0.76 | 0.99 | 1.32 | 1.34 | 1.21 | 1.37 | 1.32 |
| Korean | 1.43 | 0.56 | 1.03 | 1.59 | 1.37 | 1.14 | 1.24 | 1.59 |
| Latin | 3.13 | 2.83 | 3.03 | 3.01 | 2.91 | 1.78 | 2.21 | 3.01 |
| Norwegian | 3.03 | 2.59 | 2.91 | 2.96 | 2.78 | 1.72 | 2.08 | 2.96 |
| Persian | 2.30 | 0.87 | 1.36 | 2.76 | 1.67 | 1.48 | 1.77 | 2.76 |
| Polish | 2.83 | 2.06 | 2.56 | 2.89 | 2.58 | 1.68 | 2.02 | 2.89 |
| Portuguese | 3.36 | 2.65 | 3.23 | 3.50 | 3.15 | 1.71 | 2.08 | 3.50 |
| Russian | 2.81 | 0.98 | 1.97 | 2.87 | 2.46 | 1.57 | 1.85 | 2.87 |
| Sanskrit | 1.61 | 0.60 | 0.93 | 1.82 | 1.03 | 1.24 | 1.47 | 1.82 |
| Spanish | 3.54 | 2.78 | 3.40 | 3.71 | 3.32 | 1.73 | 2.10 | 3.71 |
| Swedish | 3.16 | 2.44 | 2.85 | 3.02 | 2.79 | 1.79 | 2.20 | 3.02 |
| Turkish | 2.50 | 2.08 | 2.52 | 3.02 | 2.77 | 1.75 | 2.20 | 3.02 |
| Vietnamese | 2.47 | 1.40 | 2.23 | 2.97 | 2.87 | 1.52 | 1.88 | 2.97 |

Figure 1: Average characters per token. Lower values indicate lower compression efficiency. Our bvv241 variants prioritize character-level granularity.

The bvv241-nemo variant is provided as a demonstration of the flexibility of the proposed methodology. While the underlying Unicode-centric mapping scheme is applicable to any tokenizer, bvv241-nemo specifically replicates the vocabulary, token structure, and exact token enumeration of the SOTA Mistral Nemo tokenizer, using its public vocabulary files as a template. This showcases how the approach can serve as a drop-in replacement or augmentation for existing BPE or unigram schemes.

Furthermore, since all token indices and their corresponding Unicode/codepoint representations are deterministic and exhaustively specified, this methodology enables the direct creation of a pre-initialized embedding matrix of shape vocab_size × n_embed. Such a matrix can be either used as a fixed (non-trainable) embedding or as a high-quality cold start, reducing the need for extensive embedding pre-training. Combining strict Unicode mapping with multi-character tokens, the bvv241 family achieves highly stable performance for Asian, Arabic, and underrepresented scripts—where differences from SOTA are minimal. For Indo-European (Latin and partially Cyrillic) languages compression is less aggressive.

## 3.3 MODEL ARCHITECTURE

We employ a standard decoder-only Transformer architecture, similar to GPT-2. All models use a hidden dimension d_model=1024, n_layer = 16, multi-head attention, n_head = 32, rotary embeddings, and GELU activations. The key modification is that the token embedding layer (nn.Embedding) has its weights frozen to the precomputed visual matrix (labeled as 'Model UNI_GLIF', total parameters: 335.8M, trainable parameters: 268.7M, frozen parameters: 67.1M). For comparison, we train baseline model with identical architectures but with a standard, randomly initialized and trainable embedding layer (labeled as 'Model unfrozen', total parameters: 335.8M, trainable parameters: 335.8M, frozen parameters: 0.0M). For ablation study: model with 16-dimensional binary frozen embeddings (labeled as 'Model_16_bin', total parameters: 269.7M, trainable parameters: 268.7M frozen parameters: 1.0M, d_model=1024, n_embed=16), model with 16-dimensional float frozen embeddings (labeled as 'Model_16_float', total parameters: 269.7M, trainable parameters: 268.7M frozen parameters: 1.0M, d_model=1024, n_embed=16). The 16-dimensional vector is projected into the model's 1024-dimensional space via repetition (repeat_interleave(64, dim=-1)), creating a simple, non-trainable mapping.). The model does not use a tied input/output embedding design; the output projection head is trained independently.

# 4 EXPERIMENTAL SETUP

## 4.1 TRAINING DATASETS

The training corpus comprises a multilingual mix including subsets of Wikipedia (en, ru, zh), and SFT datasets (10% of iterations) SQuAD-v2, TriviaQA, HotpotQA, NQ-Open, BoolQ, CommonsenseQA, and ARC: approximately 4B tokens (train 4'121'073'286 tokens + SFT train 419'136 tokens, validation 457'897'032 tokens).

## 4.2 BASELINES AND COMPARISONS

Baselines include classic transformer LMs with trainable embedding layers matched for architecture, parameter count, and tokenizer coverage. Our primary comparisons are between models with 'Frozen' embeddings and their 'Trainable' counterparts (e.g. 'Model unfrozen' vs. 'Model UNI_GLIF'), which share identical architecture and training data and tokenizer 'bvv241-2-3' (vocab size: 65536).

## 4.3 EVALUATION

We report standard language model metrics:

- Train and validation loss/perplexity,

- MMLU,

- ARC and CommonsenseQA.

All metrics are computed on held-out test sets.

## 4.4 IMPLEMENTATION DETAILS

Training was performed on H100 80 Gb with batch sizes up to 18 and block sizes of 1024. Evaluation interval was 500 (each 500 iterations MMLU, ARC, C-SENSE metrics recalculated), gradient accumulation steps parameter was 250, total iterations was set to 400'000 due to strict computation and time limits (1.7 epochs). The final model checkpoint after 400'000 iterations was used for final evaluation.

# 5 RESULTS

Before presenting our results, it is crucial to frame their context. This study does not aim to achieve state-of-the-art performance on benchmarks like MMLU. Our models are intentionally constrained in scale (up to 0.3B parameters) and trained on a small dataset. The primary objective is to demonstrate the feasibility of learning with non-semantic embeddings and to compare the learning dynamics between our proposed frozen-embedding models and traditional trainable-embedding baselines under identical conditions. The following metrics should therefore be interpreted as evidence of emergent abstraction, not as a challenge to large-scale models.

## 5.1 MODEL CONVERGENCE

A primary concern was whether a model with non-semantic, frozen embeddings could converge at all. Figure 2 and Figure 3 show the training loss curves and MMLU, ARC-e performance for a 0.3B frozen-embedding model('Model UNI_GLIF') and its trainable-embedding counterpart ('Model unfrozen'). Both models exhibit stable convergence profiles, with nearly identical loss trajectories.

Of particular note is the convergence of models with 16-dimensional binary embeddings (Model_16_bit) and 16-dimensional float embeddings (Model_16_float), which effectively carried information only about the token index to distinguish one token from another, without any information about the visual representation of the token text.

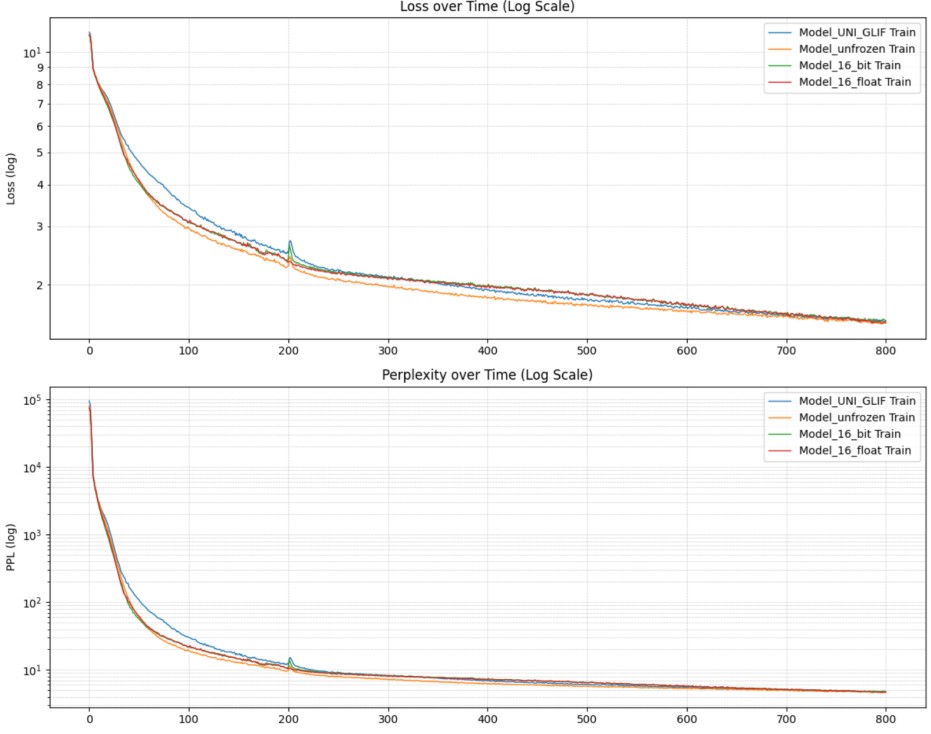

Figure 2: Learning curves for frozen vs. trainable embedding models.

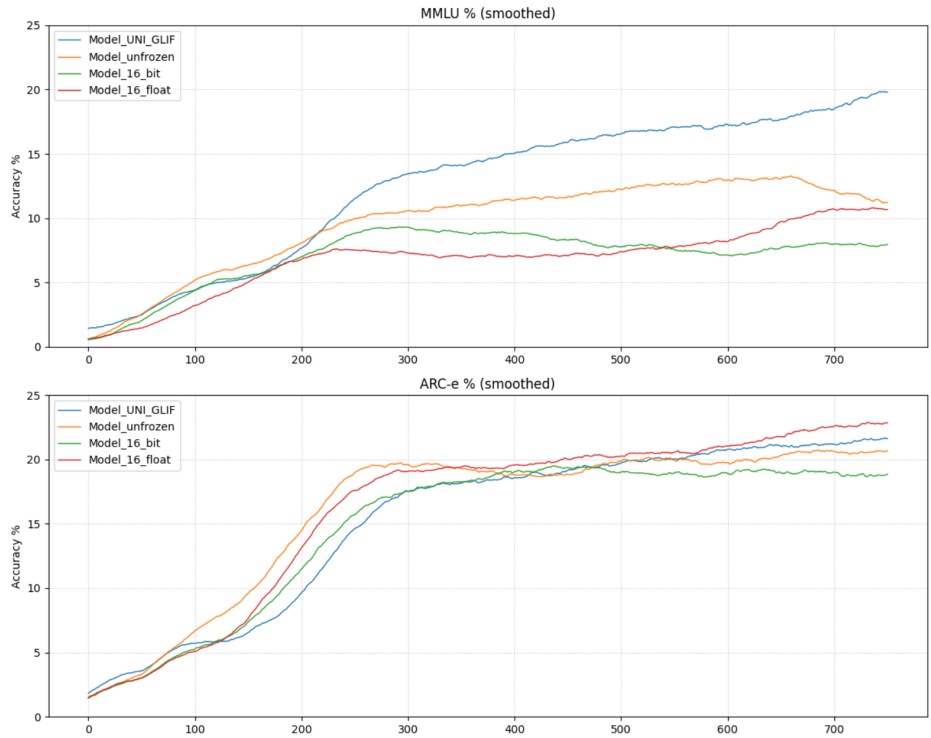

Figure 3: MMLU and ARC-e curves for frozen vs. trainable embedding models.

## 5.2 PERFORMANCE OF FROZEN VS. TRAINABLE EMBEDDING AND SOTA MODELS

Our primary finding challenges the conventional wisdom that trainable, semantically-rich embeddings are necessary for high-level reasoning. As shown in Figure 4 and Figure 5, while our frozen-embedding models demonstrate robust convergence across all tasks, they show a striking and consistent performance advantage on the MMLU reasoning benchmark compared to their architecturally identical counterparts with trainable embeddings. We also conducted a comparison with state-of-the-art (SOTA) models SmolLM (Allal et al., 2024) and SmolLM2 (Allal et al., 2025) (360M and 135M) that have a comparable number of parameters.

For instance, 'Model UNI_GLIF' (0.3B params, frozen embeddings) achieves an MMLU score of 23.81, whereas 'Model unfrozen' (0.3B params, trainable embeddings) scores only 12.54 — a nearly 2x performance gap.

We hypothesize this phenomenon, which we term "representational interference," stems from the dual burden placed on trainable embeddings. A trainable embedding layer must simultaneously learn low-level token identification task in terms of vector representation and high-level semantic abstractions within the same parameter space. This creates a conflict during optimization, leading to a suboptimal representation that is neither a perfect structural encoder nor a pure semantic vector.

In contrast, our frozen visual embeddings provide a stable, information-rich structural foundation. By offloading the task of representing token form to a fixed, deterministic layer, we free the Transformer's attention and feed-forward layers to focus exclusively on their core competency: composing these structural vector primitives into high-level meaning. The model does not waste capacity learning what a token looks like; it only learns what to do with it.

This hypothesis is further supported by the t-SNE visualizations of the embedding spaces below.

| MMLU [clinical_knowledge] | 24.2 | 17.1 | 11.0 | 16.1 | 18.1 | 25.7 | 10.9 | 21.1 |
|---|---|---|---|---|---|---|---|---|
| MMLU [computer_security] | 23.6 | 12.7 | 11.0 | 11.2 | 26.0 | 22.0 | 18.0 | 26.0 |
| MMLU [formal_logic] | 23.2 | 11.5 | 9.8 | 12.1 | 20.6 | 30.2 | 17.5 | 21.4 |
| MMLU [global_facts] | 24.6 | 8.9 | 7.9 | 8.0 | 17.0 | 16.0 | 10.0 | 29.0 |
| MMLU [high_school_computer_science] | 22.5 | 11.8 | 12.6 | 11.1 | 21.0 | 26.0 | 12.0 | 26.0 |
| MMLU [high_school_geography] | 24.7 | 12.7 | 11.3 | 13.7 | 11.6 | 17.7 | 14.7 | 26.8 |
| MMLU [high_school_government_and_politics] | 23.8 | 10.2 | 10.6 | 9.7 | 18.6 | 20.7 | 12.4 | 29.5 |
| MMLU [high_school_macroeconomics] | 25.2 | 14.5 | 10.9 | 14.3 | 11.0 | 22.3 | 11.3 | 24.9 |
| MMLU [high_school_psychology] | 24.2 | 15.3 | 12.0 | 14.7 | 17.1 | 20.9 | 18.0 | 26.8 |
| MMLU [high_school_us_history] | 22.1 | 8.5 | 10.2 | 8.1 | 27.0 | 24.0 | 20.6 | 22.6 |
| MMLU [high_school_world_history] | 22.6 | 10.5 | 9.9 | 8.5 | 29.1 | 23.6 | 20.7 | 24.9 |
| MMLU [human_aging] | 24.2 | 12.7 | 14.7 | 13.1 | 24.2 | 29.1 | 16.1 | 23.3 |
| MMLU [human_sexuality] | 24.4 | 14.1 | 11.7 | 14.3 | 22.1 | 22.1 | 14.5 | 25.2 |
| MMLU [jurisprudence] | 23.0 | 11.6 | 11.8 | 12.8 | 20.4 | 21.3 | 15.7 | 27.8 |
| MMLU [management] | 24.4 | 10.3 | 10.2 | 10.9 | 21.4 | 14.6 | 11.7 | 28.2 |
| MMLU [marketing] | 25.3 | 16.3 | 11.9 | 18.1 | 20.9 | 24.4 | 9.4 | 20.9 |
| MMLU [medical_genetics] | 23.7 | 15.5 | 12.4 | 13.6 | 27.0 | 35.0 | 14.0 | 25.0 |
| MMLU [moral_disputes] | 24.7 | 12.6 | 10.3 | 15.6 | 26.9 | 24.3 | 12.1 | 22.2 |
| MMLU [philosophy] | 23.1 | 14.0 | 10.6 | 16.2 | 18.3 | 17.0 | 11.9 | 28.6 |
| MMLU [professional_accounting] | 25.4 | 12.9 | 12.8 | 9.6 | 18.8 | 17.4 | 8.2 | 24.5 |
| MMLU [professional_medicine] | 23.9 | 11.7 | 13.0 | 12.1 | 18.8 | 17.6 | 14.7 | 32.7 |
| MMLU [public_relations] | 24.9 | 14.4 | 12.9 | 10.2 | 18.2 | 21.8 | 11.8 | 28.2 |
| MMLU [security_studies] | 18.4 | 9.1 | 8.1 | 9.3 | 18.8 | 13.9 | 26.5 | 22.9 |
| MMLU [us_foreign_policy] | 25.6 | 11.7 | 12.4 | 7.9 | 26.0 | 30.0 | 18.0 | 32.0 |
| | Model UNI_GLIF (frozen) | Model unfrozen (baseline) | Model_16 bit (frozen) | Model_16 float (frozen) | SmolLM 135M | SmolLM 360M | SmolLM2 135M | SmolLM2 360M |

Figure 4: Performance Comparison: MMLU sub tasks with score greater than 25%.

| C-SENSE | 19.75 | 19.75 | 19.16 | 19.89 | 32.70 | 35.30 | 33.90 |
|---|---|---|---|---|---|---|---|
| MMLU | 23.81 | 12.54 | 10.93 | 11.35 | 28.30 | 30.60 | 29.30 |
| | Model UNI_GLIF (frozen) | Model unfrozen (baseline) | Model_16 bit (frozen) | Model_16 float (frozen) | SmolLM 135M | SmolLM- 360M | SmolLM2 135M |

Figure 5: Performance Comparison: Frozen vs. Trainable Embedding and SOTA SmolLM Models.

It is critical to contextualize our baseline's performance against the SOTA models used in our comparisons. The SmolLM models (28% MMLU) were trained on up to 600B tokens (smollm-corpus), whereas our experiments were deliberately constrained to a 4B token dataset—a 150x difference in compute and data. Our goal was not to match SOTA but to ensure a perfectly fair comparison between our proposed method and the baseline under identical, resource-constrained conditions (the only difference was the frozen/unfrozen embedding layer).

Under these severe data constraints, it is the conventional trainable baseline (Model_unfrozen, 12.5% MMLU) that dramatically underperforms. In contrast, our frozen-embedding model (Model_UNI_GLIF, 23.8% MMLU) achieves a score comparable to established models like GPT-2 137M (25.8%-26.29% MMLU).

## 5.3 EMERGENT SEMANTICS VISUALIZATION OF INPUT TOKEN EMBEDDINGS AND DEEP TRANSFORMER LAYERS

Our fixed visual embedding vectors naturally cluster by surface/formal features (e.g., length), rather than by meaning (Figure 7). In contrast, trainable embeddings demonstrate soft semantic grouping, though the distributed latent space remains globally unstructured (Figure 6).

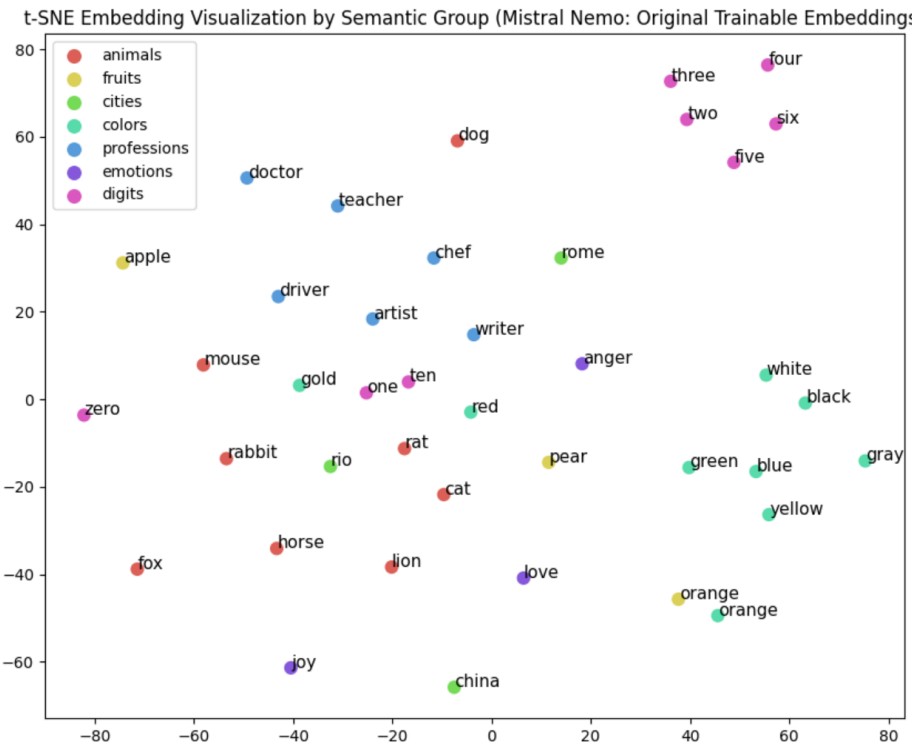

Figure 6: t-SNE visualization of input token embeddings. In the trainable Mistral Nemo embeddings, semantic groups (e.g., numbers, professions, animals) tend to lie closer to one another, yet, globally, the point cloud remains mostly uniform.

The following part presents a method for analyzing internal representations of a language model through extraction and visualization of activations from various neural network layers. The study examined 70 words organized into seven semantic groups (royalty, animals, cities, actions, objects, feelings, professions). Individual words were fed as input to the model, and vector representations were extracted using forward hooks mechanism from key architectural components: input embeddings, attention projections (v_proj and o_proj), and transformer block outputs across all layers of the model.

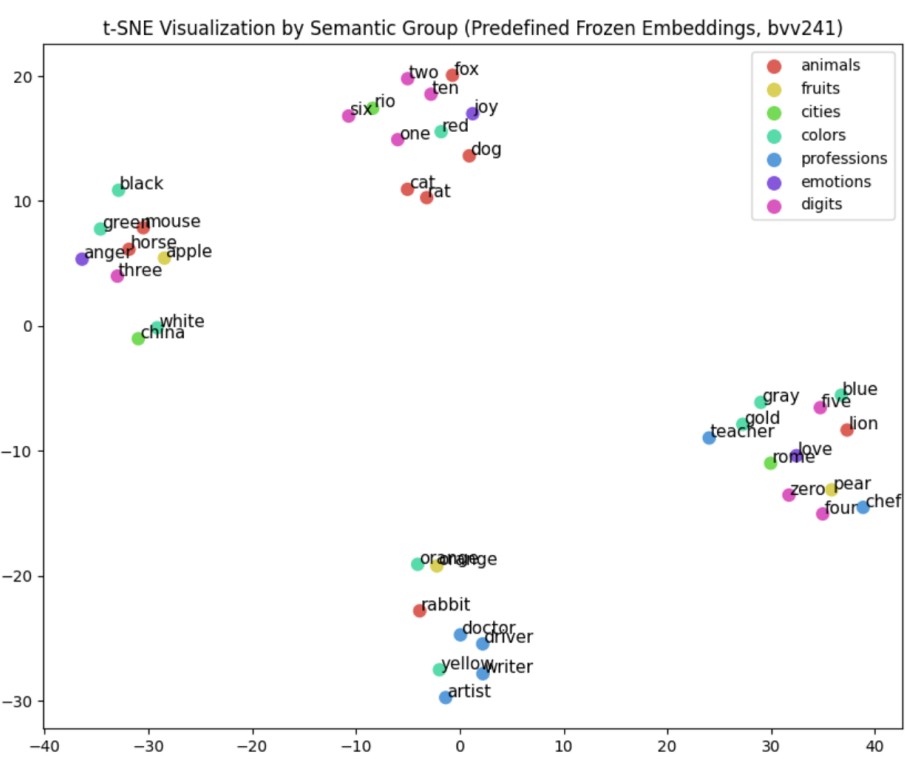

Figure 7: t-SNE visualization of the frozen visual embeddings. The prominent clustering directly corresponds to token length. This phenomenon is a direct artifact of our embedding generation process: multi-character tokens, when rendered and resized to a fixed-resolution bitmap, produce images with higher average pixel density. As Principal Component Analysis (PCA) is designed to capture axes of maximum variance, it naturally identifies this structural feature as a dominant component. The resulting clusters are therefore based on a physical token property (length/density), not semantic content, reinforcing the conclusion that semantic understanding must emerge entirely within the subsequent Transformer layers.

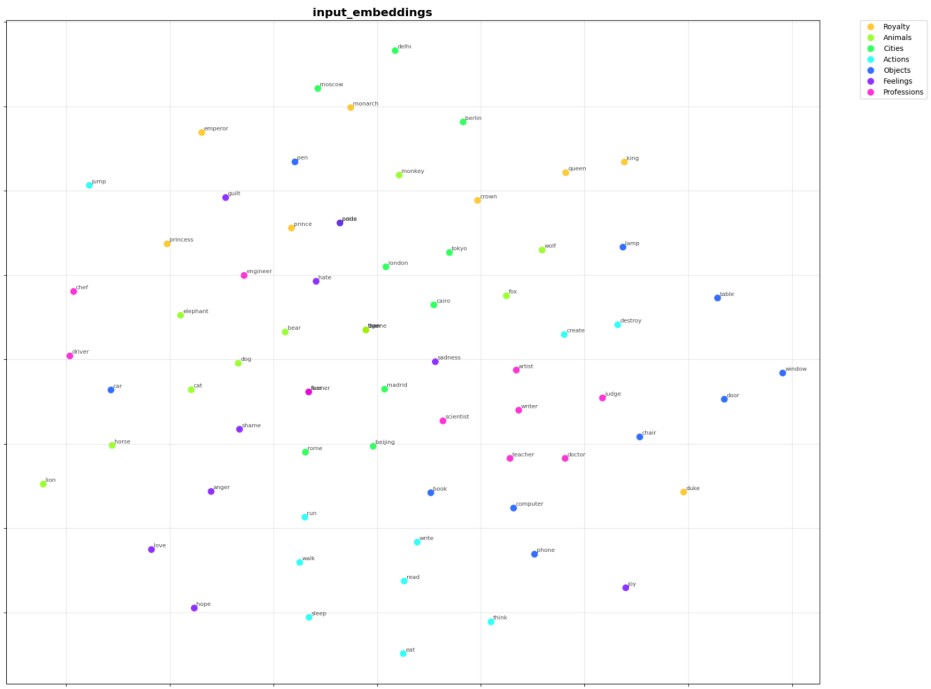

Figure 8: t-SNE visualization of input token embeddings for baseline ('Model unfrozen'). In this trainable embeddings, semantic groups (e.g., numbers, professions, animals) tend to lie closer to one another and clustering.

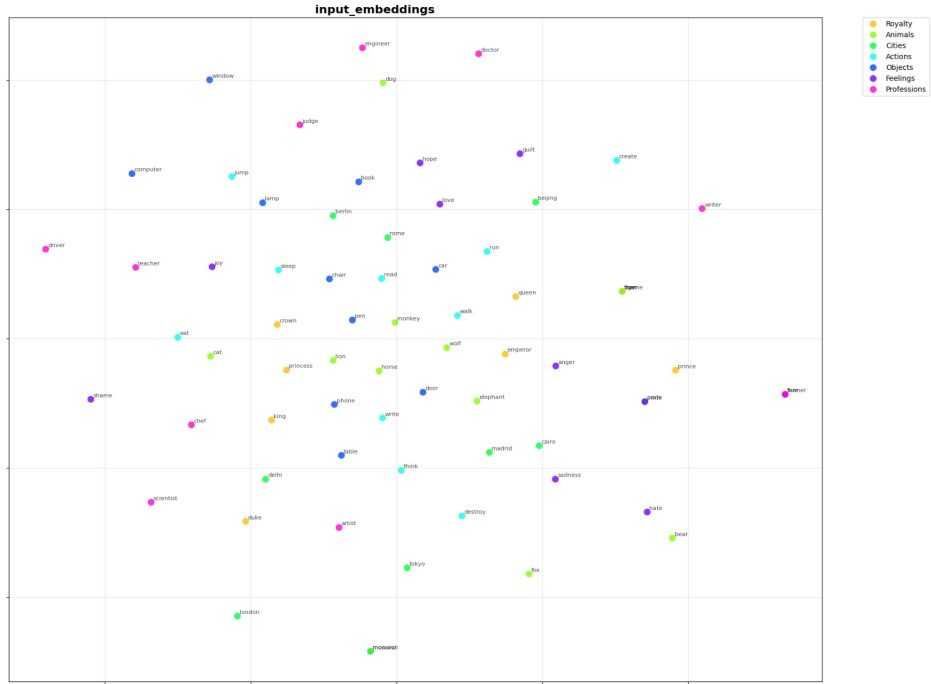

Figure 9: t-SNE visualization of frozen input token embeddings for 'Model_UNI_GLIF'. In the this frozen embeddings, semantic groups (e.g., numbers, professions, animals) located randomly and without clustering.

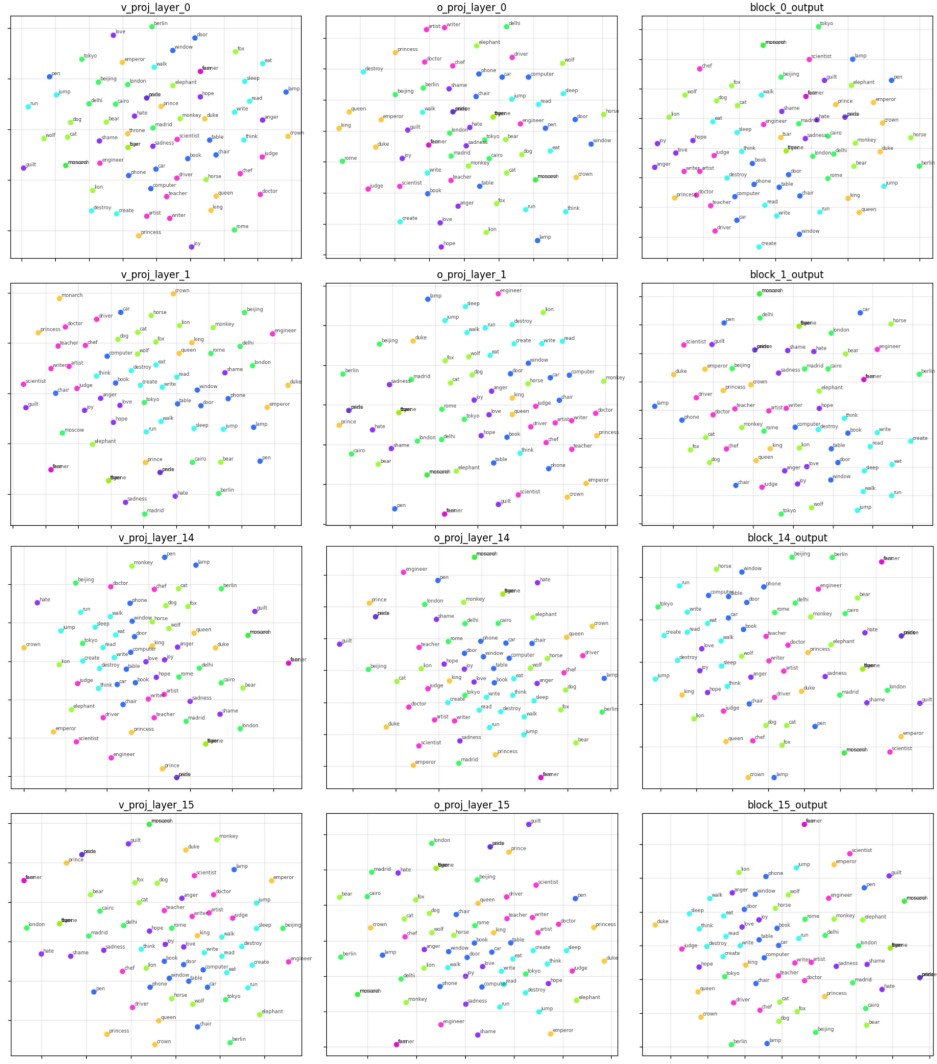

Figure 10: t-SNE visualization of first two (layer 0 - layer 1) and the last two (layer 14 - layer 15) attention projections (v_proj and o_proj), and transformer block outputs for baseline ('Model unfrozen'). In this picture semantic groups tend to clustering.

The collected high-dimensional vectors were normalized using StandardScaler, followed by t-SNE algorithm application for dimensionality reduction to a two-dimensional space for visualization purposes. This approach enabled tracking the evolution of semantic representations as information propagates through the model layers. The analysis reveals how initially dispersed tokens gradually organize into semantically related clusters in deeper network layers, demonstrating the model's capability for hierarchical extraction and structuring of semantic features. This approach provides a tool for interpreting the "black box" of transformer models and understanding the mechanisms of semantic representation formation during natural language processing.

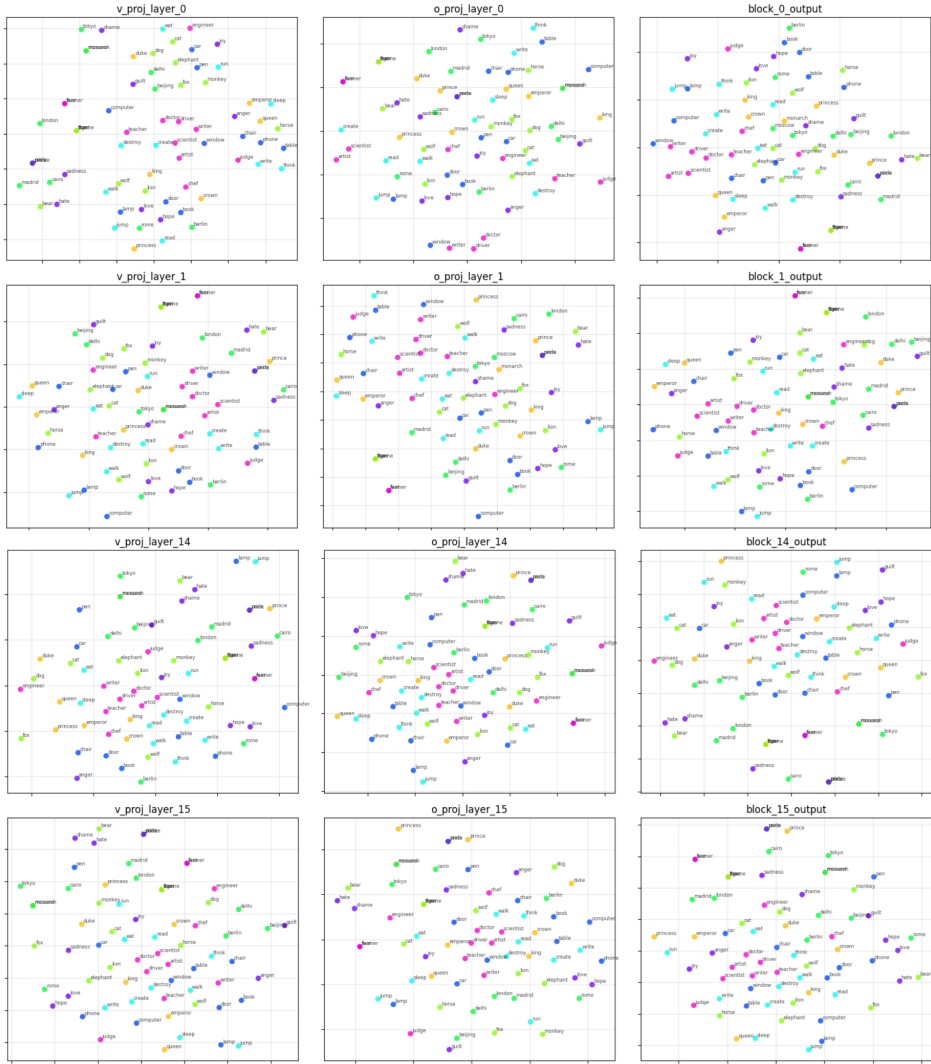

Figure 11: t-SNE visualization of first two (layer 0 - layer 1) and the last two (layer 14 - layer 15) attention projections (v_proj and o_proj), and transformer block outputs for 'Model_UNI_GLIF' with frozen input embedding layer. In this picture semantic groups (e.g., numbers, professions, animals) tend to make clusters after frozen embedding layer demonstrating emergence of semantic structure. This visualization provides direct evidence that semantic clustering, absent in the frozen input, is formed by the model's compositional blocks.

### 5.4 ABLATION STUDY: THE ROLE OF EMBEDDING INITIALIZATION

This ablation study compares the performance of traditional trainable embeddings, frozen embeddings initialized with Unicode-based visual representations, and frozen embeddings initialized with random black-and-white bitmaps, which serve as a randomized baseline.

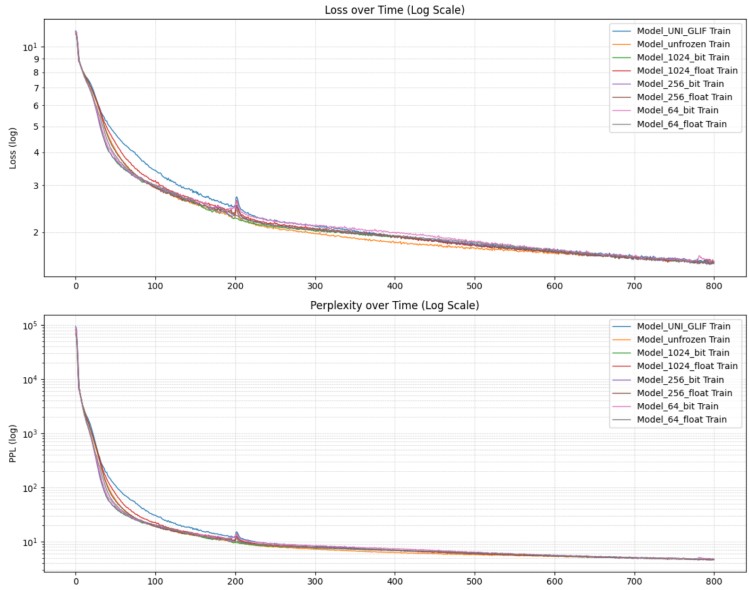

Figure 12: Learning curves for the baseline model 'Model unfrozen' vs frozen nn.Embedding layer model 'Model UNI_GLIF' and randomly initialized frozen nn.Embedding layer models

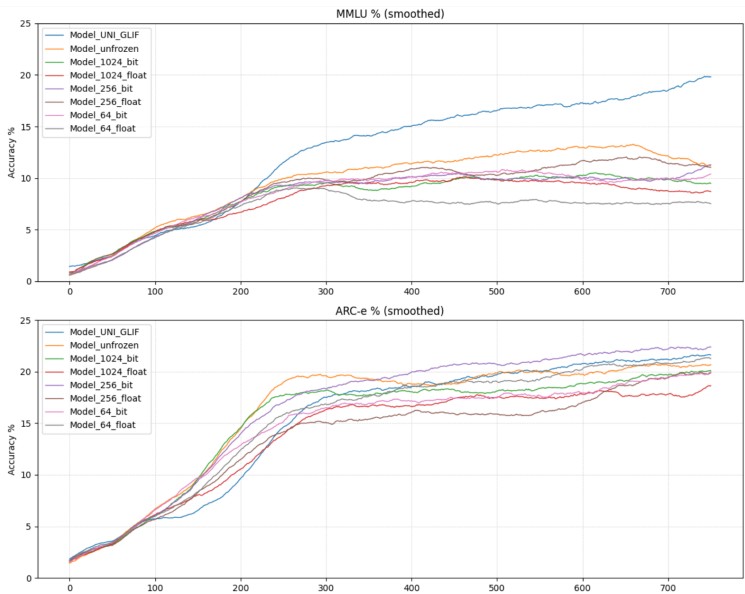

Figure 13: MMLU and ARC-e performance for the baseline model 'Model unfrozen' vs frozen nn.Embedding layer model 'Model UNI_GLIF' and randomly initialized frozen nn.Embedding layer models

This ablation study reveals an insight: even when frozen embeddings are initialized from random noise, the model still converges (Figure 12, Figure 13, Figure 14). This finding suggests that the semantic structures

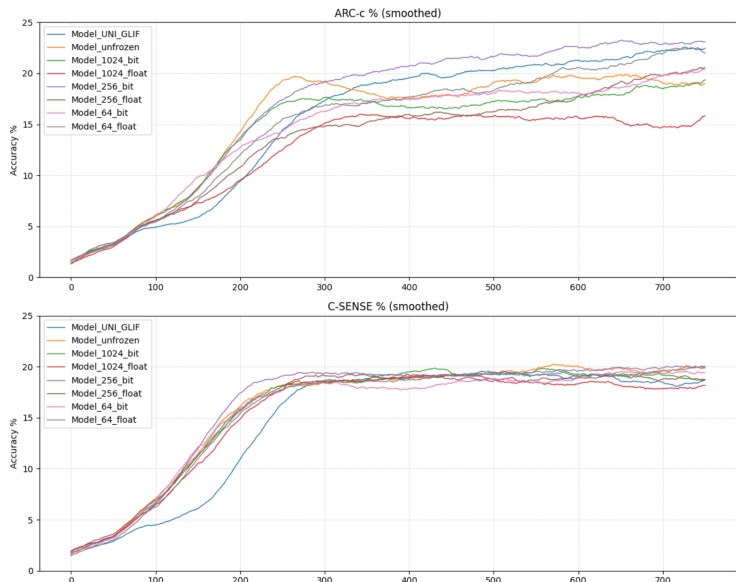

Figure 14: ARC-c and C-SENSE performance for the baseline model 'Model unfrozen' vs frozen nn.Embedding layer model 'Model UNI_GLIF' and randomly initialized frozen nn.Embedding layer models

observed in trainable embeddings are not a prerequisite for learning, but rather an emergent artifact. We hypothesize that this phenomenon stems from a trainable embedding layer being implicitly forced to solve two conflicting tasks: 1) Token Discrimination, the primary goal of assigning a unique vector to each token, and 2) Semantic Proximity, the secondary goal of clustering vectors for contextually similar tokens. These objectives are often at odds. In essence, during backpropagation, gradients from high-level contextual errors "imprint" semantic relationships onto the embedding vectors, forcing them to serve as a low-dimensional projection surface for knowledge learned by the entire network. Our proposed frozen-embedding approach resolves this conflict by decoupling the tasks. By providing a stable, unique identifier for each token, the fixed vectors (whether visually or randomly initialized) perfectly satisfy the discrimination objective. This liberates the Transformer's compositional layers—the attention and feed-forward networks—to exclusively manage the learning of semantic relationships. We posit that this separation of concerns reduces optimization conflicts and allows the architecture to learn more efficiently, as evidenced by the superior performance on reasoning benchmarks.

## 6 DISCUSSION

Our findings suggest that the role of input embeddings in Transformers may be widely misunderstood. Meaning appears not to be a property of the embeddings, but a process enacted by the architecture.

### 6.1 RE-EVALUATING EMBEDDINGS: FROM MEANING CONTAINERS TO STRUCTURAL PRIMITIVES

The success of our frozen-embedding models, particularly on MMLU, challenges the view of embeddings as "meaning vectors." Instead, they can be seen as "structural primitives." The "representational interference" hypothesis suggests a fundamental inefficiency in standard LLMs. By tasking the embedding layer with both structural and semantic learning, we may be creating an optimization bottleneck that harms downstream reasoning. Forcing the model to learn, for instance, that the token "king" is semantically close to "queen" but structurally dissimilar from "monarch" within the same vector space is a difficult, potentially conflicting task.

Our method resolves this by offloading the structural representation to a fixed, deterministic layer. The model doesn't waste capacity learning what a token looks like; it only learns what to do with its form in context. This may be particularly advantageous for tasks requiring orthographic or morphological awareness, which could be implicitly tested in MMLU's diverse sub-tasks.

This suggests a hidden inefficiency in standard models. Consider a task requiring orthographic reasoning (e.g., "How many 'e's in 'strawberries'?"). If "strawberries" is a single token, its semantic embedding contains no direct information about its character composition. The model must perform a costly "internal detokenization," using its deep layers to implicitly reconstruct the word's structure from memory. Our approach mitigates this by providing structural information at the input layer, potentially making the model more computationally efficient for a class of reasoning tasks. The performance gap on MMLU suggests that decoupling structural representation from semantic composition is a more effective learning strategy, especially under constrained data and compute.

## 6.2 IMPLICATIONS FOR ARCHITECTURE AND EFFICIENCY

This paradigm opens several avenues for future architectures:

- **Modular and Standardized Embeddings:** A universal, algorithmically-derived visual embedding matrix could become a standard component, allowing researchers to swap out different Transformer backbones for direct comparison without confounding factors from different embedding initializations.

- **Efficient MoE and Model Fusion:** Mixture-of-Experts (MoE) models, which typically shard their MLP layers, could additionally benefit from a shared frozen embedding layer, reducing total parameter count and simplifying the routing logic by providing a common input space for all experts.

- **Improved Cross-Lingual Transfer:** A single Unicode-based visual embedding space is inherently multilingual. This could provide a more robust foundation for zero-shot cross-lingual transfer, as the model learns to operate on visual script features common to many languages.

## 6.3 LIMITATIONS AND FUTURE WORK

This study was intentionally constrained in scale. Future work should test whether these findings hold for models at the 10B+ parameter scale. Additionally, while our method covers the entire Unicode text space, its generalization to truly novel modalities (e.g., mathematical formulas, chemical structures) remains an open question.

## 7 CONCLUSION

We have demonstrated that transformer LMs with frozen, purely visual Unicode-based embeddings can learn rich linguistic and logical abstraction. Semantics in neural LMs is not a property of the embedding matrix, but an emergent phenomenon of the model's layered composition and self-attention. This insight opens new avenues for model standardization, modularity, and efficient scaling in multi-lingual and multi-domain NLP.

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

## 8   APPENDIX A: Model Parameters and Vocabulary Statistics

Table 1: Model Parameters and Vocabulary Statistics. Overview of models used in experiments. 'Frozen' models use our fixed visual embeddings. 'Trainable' models are baselines with standard, randomly initialized, trainable embeddings. All other architectural parameters (d_model=1024, layers, etc.) are held constant between frozen and trainable embeddings mode.

| Model Name | Parameters and Vocab Size | Embedding Type |
|---|---|---|
| Model_UNI_GLIF | 0.3 B, 65536 | Frozen (Visual Unicode) |
| Model_unfrozen | 0.3 B, 65536 | Trainable |
| Model_16_bit | 0.3 B, 65536 | Frozen (Sequential 16-bit) |
| Model_16_float | 0.3 B, 65536 | Frozen (Sequential 16-float PCA) |
| Model_64_bit | 0.3 B, 65536 | Frozen (Random 64-bit) |
| Model_64_float | 0.3 B, 65536 | Frozen (Random 64-float PCA) |
| Model_256_bit | 0.3 B, 65536 | Frozen (Random 256-bit) |
| Model_256_float | 0.3 B, 65536 | Frozen (Random 256-float PCA) |
| Model_1024_bit | 0.3 B, 65536 | Frozen (Random 1024-bit) |
| Model_1024_float | 0.3 B, 65536 | Frozen (Random 1024-float PCA) |

## 9   APPENDIX B: Training loss and perplexity data

Table 2: Training loss and perplexity data

| Step | Model UNI_GLIF Loss | Model UNI_GLIF PPL | Model unfrozen Loss | Model unfrozen PPL | Model 16_bit Loss | Model 16_bit PPL | Model 16_float Loss | Model 16_float PPL |
|---|---|---|---|---|---|---|---|---|
| 0 | 11.46 | 95123.07 | 11.30 | 80778.27 | 11.20 | 73204.74 | 11.23 | 75269.51 |
| 050000 | 3.42 | 30.51 | 2.96 | 19.21 | 3.10 | 22.17 | 3.08 | 21.69 |
| 100000 | 2.50 | 12.13 | 2.28 | 9.75 | 2.41 | 11.17 | 2.35 | 10.47 |
| 150000 | 2.10 | 8.20 | 1.97 | 7.19 | 2.08 | 8.04 | 2.08 | 7.98 |
| 200000 | 1.93 | 6.89 | 1.82 | 6.18 | 1.99 | 7.30 | 1.98 | 7.22 |
| 250000 | 1.81 | 6.10 | 1.74 | 5.72 | 1.88 | 6.53 | 1.86 | 6.45 |
| 300000 | 1.72 | 5.57 | 1.67 | 5.30 | 1.74 | 5.70 | 1.75 | 5.74 |
| 350000 | 1.62 | 5.05 | 1.62 | 5.08 | 1.63 | 5.12 | 1.62 | 5.07 |
| 400000 | 1.54 | 4.68 | 1.53 | 4.64 | 1.56 | 4.78 | 1.55 | 4.73 |