# OpenReview forum: "Growing Transformers: Modular Composition and Layer-wise Expansion on a Frozen Substrate"
_ICML.cc/2026/Conference — Submitted to ICML 2026_

### Official Review · Reviewer_C1kq · 2026-03-09

**Soundness:** 2
**Presentation:** 1
**Significance:** 1
**Originality:** 1
**Overall Recommendation:** 2
**Confidence:** 4

**Summary:**

The paper studies progressive layer-wise growth of decoder-only Transformers on a frozen embedding substrate. The procedure: train a single block, freeze it, stack a new randomly initialized block on top, repeat. Only the newest block(s) and the LM head are trained at each stage. The embedding layer is kept frozen throughout, either as a standard Unicode mapping or a 16-dimensional binary token-ID vector. Constructive growth on a frozen substrate achieves competitive MMLU and reasonable SQuAD EM relative to monolithic training, with reduced peak trainable parameters. The paper is a systems-level feasibility demonstration that offers limited novelty over established work on progressive depth growth.

**Compliance With Llm Reviewing Policy:**

Affirmed.

**Final Justification:**

Rebuttal partially addressed my concerns, insufficient comparisons with prior baselines limit my maximal score to reject.

**Key Questions For Authors:**

1. How does your method compare to G_stack (Du et al.), which duplicates trained layers rather than randomly initializing new ones?

2. Can you provide any formal analysis of the representational capacity of a model with frozen lower blocks?

3. Can you provide results on a purely unsupervised corpus to isolate the growth procedure from the instruction data mixture?

**Limitations:**

Discussed adequately, though the paper does not address the fundamental limitation that freezing lower layers permanently constrains the model's ability to update early representations, which may worsen at larger scales.

**Strengths And Weaknesses:**

**Strengths:**

The frozen-embedding substrate is the most interesting aspect. The demonstration that a 16-dimensional binary token-ID embedding supports competitive constructive growth is an unexpected result. The t-SNE visualizations (Figure 7) show semantic structure emerging in deeper layers despite non-semantic input representations.

The fixed trainable-parameter budget framing is well-motivated and the controlled comparative study is carefully designed, with ablations isolating the frozen-embedding effect from the growth procedure.

**Weaknesses:**

Limited Novelty: Progressive depth growth for Transformers is thoroughly explored. "Efficient Training of BERT by Progressively Stacking" (Gong et al.) introduced progressive stacking for BERT. "Net2Net: Accelerating Learning via Knowledge Transfer" (Chen et al.) formalized function-preserving depth expansion. "Progressively Stacking 2.0" (Yang et al.) extended this with multi-stage layerwise training where only new layers are updated and lower layers are frozen. "Stacking Your Transformers" (Du et al.) conducted a comprehensive study of stacking operators for LLM pre-training up to 7B parameters with 750B tokens. "Learning to Grow Pretrained Models for Efficient Transformer Training" (Wang et al.) learned linear growth operators (LiGO) subsuming stacking and copying as special cases. The present paper's contribution, freezing lower blocks and training only the top, is a simplification of these methods.

No Theoretical Analysis: The paper provides no formal analysis of why constructive growth on a frozen substrate should work, when it might fail, or how frozen lower blocks affect representational capacity. Prior work has at least provided intuitions grounded in function-preserving transformations or optimization landscape analysis.

Missing Baselines: The paper does not compare against the standard approach of duplicating trained layers to initialize new ones (G_stack). The proposed method initializes new blocks randomly, which is known to be suboptimal relative to function-preserving or duplication-based growth. This is a significant gap.

---

> ### Author Rebuttal · Authors · 2026-03-30
>
> Thank you for the detailed review. We agree that the related-work coverage should be expanded, and the papers you cite on progressive stacking / Net2Net-style growth / LiGO / stacking studies for LLM pretraining should have been discussed explicitly.
>
> That said, we believe the paper’s target regime is narrower than “generic progressive depth growth.” The intended contribution is not simply that depth can be grown. It is a fixed-local-budget, non-monolithic training regime in which (i) the bottom interface remains fixed, including in the extreme case of a frozen 16-d binary token-ID substrate, (ii) all previously trained dense blocks remain frozen, and (iii) the local trainable set / optimizer-state footprint is kept approximately constant throughout training, with LoRA used only as a budget-preserving global adjustment. No stage trains the full dense model end-to-end. This fixed local optimizer-state budget is the main object of study and is the sense in which we view the paper as distinct from generic progressive stacking work.
>
> Regarding duplication-based growth / G_stack: we agree this is a useful missing baseline. Our choice of random initialization for newly added blocks was deliberate because the question here is stricter: can continued learning proceed above a fixed bottom interface and a frozen dense lower stack even without function-preserving copying? We do not claim that random initialization is the best growth initializer; a duplication-based initializer may well improve absolute perplexity. The point here is feasibility of continued learning under the fixed-local-budget regime, not optimality of the initializer. We agree that adding G_stack-style comparisons would strengthen a future revision.
>
> Regarding theory: the paper makes empirical feasibility/training-regime claims rather than formal claims about representational capacity or optimization guarantees. We agree that the mechanistic language should therefore be conservative, and we are happy to tone down any wording that reads like a formal claim about why the method must work or when it must fail. A formal analysis of capacity with frozen lower blocks would be valuable future work, but is outside the present paper.
>
> Regarding data mixture / purely unsupervised isolation: we agree that the 68.9B-token run is not a pure causal isolation of growth because stage-wise SFT mixing is present. It should be read as a long-horizon viability demonstration, not as a purely unsupervised ablation. The cleanest method comparison in the paper is instead the 9-layer controlled study, where all compared models use the same 4B-token mixture, tokenizer, architecture, and shared training protocol, and where there is no LoRA phase. We agree that a purely unsupervised version would be useful future work.
>
> Finally, on the limitation that freezing lower layers constrains the ability to update early representations: we agree this is a real limitation of the pure constructive-growth setting. However, the larger-scale method is not “permanently frozen everywhere below the top” in the sense of having no global readjustment path at all. In the extended runs we interleave global LoRA phases precisely to reintroduce limited adaptation across the entire stack while keeping the same local trainable-parameter budget. The 9-layer controlled study intentionally omits LoRA in order to isolate the strict / worst-case setting of frozen lower dense blocks only. So we agree that early dense representations are not directly updated, but the paper does provide a budget-preserving mechanism for partial global readjustment, and this distinction should have been made clearer.
>
> In this paper, the frozen 16-d binary token-ID substrate is important not as a claim that the proposed recipe is the best generic growth method, but as the strongest stress test of the fixed-interface premise: if continued growth remains viable even there, then reopening the bottom of the network is not a prerequisite for continued learning.

---

> > ### Author Rebuttal · Reviewer_C1kq · 2026-04-01
> >
> > The authors provided further, reasonable, arguments in favor of their work and have further contextualized it w.r.t prior work. However, both the evaluation and theory behind the work remain underdeveloped (especially comparisons against prior methods). I have raised my score accounting for their further arguments.

---

> > > ### Author Response · Authors · 2026-04-03
> > >
> > > Thank you again for the follow-up and for revisiting the score.
> > >
> > > We agree that the strongest missing comparison is against prior growth methods using copying / function-preserving initialization (e.g., G_stack / Net2Net / LiGO-style growth). We do not claim that random-init stacking is the best generic growth recipe.
> > >
> > > The intended claim of the paper is narrower: it studies the viability of a specific constrained regime.
> > >
> > > | Aspect | Typical prior growth work | This work |
> > > |---|---|---|
> > > | Primary aim | improve optimization / preserve function during growth | maintain a bounded active trainable / optimizer-state budget during continued learning |
> > > | Previously trained dense blocks in later stages | varies | frozen |
> > > | Later dense full-model reopening | often allowed | no dense reopening (LoRA only, when used, under the same active budget) |
> > > | Input interface | standard learned / high-capacity embeddings; usually not the object of study | fixed precomputed frozen interface (strongest stress test uses an n_embed=16 binary token-ID code) |
> > > | Explicit fixed active trainable budget as an objective | not usually central | yes |
> > >
> > > One distinction we should have made more explicit is that prior growth papers generally study expansion under conventional learned input interfaces, whereas here the input interface itself is fixed throughout growth and can be reduced to a minimal deterministic code.
> > >
> > > For clarity, the 16-d case is not a learned 1024-d embedding in disguise. Each token is represented by a fixed binary code b_t ∈ {0,1}^{16}, and the model input is a deterministic lift of that code to d_model with no trainable input projection. Thus the effective input bottleneck before the first trainable block remains 16-dimensional.
> > >
> > > Under this reading, the 16-d binary substrate is not a separate embedding proposal; it is the strongest stress test of the fixed-interface premise. The contribution we are defending is therefore the viability of bounded-active-budget non-monolithic growth above a fixed minimal interface, not the optimality of random initialization relative to duplication-based growth.
> > >
> > > We agree that adding G_stack / function-preserving baselines would materially strengthen a revision, and that our current evidence should be read as feasibility/tradeoff evidence rather than optimality evidence.
> > >
> > > We appreciate the concrete references; they have helped us sharpen the correct scope and positioning of the paper.

---

### Official Review · Reviewer_o45H · 2026-03-12

**Soundness:** 3
**Presentation:** 3
**Significance:** 2
**Originality:** 2
**Overall Recommendation:** 3
**Confidence:** 4

**Summary:**

This paper introduces a method to grow Transformer models layer-by-layer under a fixed trainable-parameter budget, significantly reducing memory requirements. The core idea is as follows: start with one block on a frozen embedding, train, freeze, then stack and train a new block—repeating to increase depth.

**Compliance With Llm Reviewing Policy:**

Affirmed.

**Key Questions For Authors:**

See weakness above.

**Limitations:**

Yes.

**Strengths And Weaknesses:**

Strength:
Training large language models under severe memory constraints is a critical practical problem. The proposed constructive growth approach offers a new solution by keeping the trainable parameter count.

Weakness:
(1) Compared to directly training the full model, is there a performance gap with the proposed method?
(2) The size of the LLM in this paper is small, and the accuracy over MMLU is generally < 25% (random), diminishing the practical value of the experiments.

---

> ### Author Rebuttal · Authors · 2026-03-30
>
> Thank you for the direct and useful comments.
>
> (1) Performance gap vs direct full-model training. Yes, in the controlled 9-layer study there is a gap relative to monolithic training: the monolithic frozen-UNICODE baseline reaches lower final PPL than the constructive model (3.71 vs 4.12). Our intended claim is therefore not zero-gap or universal superiority over full-model training. The claim is a tradeoff claim: at the same final 9-layer architecture, peak trainable parameters drop from 247.6M to 105.0M (a 57.6% reduction), i.e., bounded local optimizer-state memory rather than best absolute perplexity. We agree the paper should make this tradeoff more explicit and tone down any wording that sounds like a general performance claim.
>
> (2) Model size / MMLU. The controlled 9-layer models are modest by frontier-LM standards by design: they are the scale at which we could run matched baselines under a shared protocol. The 2.3B run is included as a feasibility demonstration of the fixed-local-budget regime, and is intentionally wide/shallow rather than benchmark-optimized. The longer-horizon 16-layer run reaches 28.92% MMLU and 19.10% SQuAD EM on a frozen 16-d binary token-ID substrate, so the paper is not uniformly below four-choice chance. We agree that these absolute scores are still modest compared to much larger modern LLMs. The practical value we intended to show is different: once continued learning can proceed above a fixed bottom interface, useful additional depth can be trained under a bounded local optimizer-state budget, in settings where dense monolithic full-model training would require a different systems regime.
>
> We will revise the framing accordingly: this is a feasibility / training-regime paper for memory-constrained scaling, not a claim of best benchmark performance.

---

> > ### Author Rebuttal · Reviewer_o45H · 2026-04-04
> >
> > Thanks for your response. Some of the concerns have been addressed. However, as some of the results of MMLU is less than 25%, these results may not demonstrate the effectiveness of the method.

---

### Official Review · Reviewer_KiB9 · 2026-03-15

**Soundness:** 2
**Presentation:** 3
**Significance:** 2
**Originality:** 4
**Overall Recommendation:** 3
**Confidence:** 4

**Summary:**

This paper proposes a layer-wise growth strategy for decoder-only Transformers, aiming to reduce the memory footprint of optimizer states during pretraining. The method involves incrementally adding new Transformer blocks to a frozen embedding substrate, training only the new blocks and the language model head at each stage. The authors claim this approach enables the training of large models with limited hardware by maintaining a fixed trainable-parameter budget.

The paper presents three experiments to validate this method. A feasibility study demonstrates growing a 2.3B-parameter model on a single GPU. A controlled study compares 248M-parameter models grown with this method against monolithically trained baselines, reporting competitive performance on MMLU. A web-scale experiment grows a 16-layer model on 68.9B tokens, showing performance improvements after an interleaved LoRA tuning stage.

However, the paper suffers from several significant scientific weaknesses that call its conclusions into question. The empirical evidence is weakened by a lack of statistical rigor (single-run experiments), confounding variables in the web-scale experiment, a very limited review of related work, and evaluation on benchmarks where model performance is low. Consequently, the claims of competitiveness with monolithic training are not fully substantiated.

**Compliance With Llm Reviewing Policy:**

Affirmed.

**Key Questions For Authors:**

1.Regarding the lack of statistical analysis: Why did the authors not perform multiple runs for their controlled experiments? Without them, how can any claims of competitiveness be justified, especially when the observed differences are so small?

2.Regarding the web-scale experiment: How can the performance gains be attributed to the growth method when the data distribution was changed concurrently? Why was a monolithic baseline not included for comparison?

3.Regarding the sparse related work: Can the authors explain the omission of highly relevant prior work in layer-wise training and model growth? How does their method compare to these existing approaches?

**Limitations:**

This paper introduces an original and interesting idea for memory-efficient model training. The concept of growing a Transformer on a frozen, non-semantic substrate is novel and thought-provoking.

However, the execution of the study has several major scientific flaws that prevent a recommendation for acceptance. The claims of competitiveness are not well-supported, a confounded experimental design in the large-scale run, and an inadequate survey of related work. While the problem is important and the idea is creative, the weaknesses in the empirical execution are too significant to overlook. The paper has potential, but it does not currently meet the bar for publication at ICML.

**Strengths And Weaknesses:**

Strengths

S1. The core concept of layer-wise growth on a frozen, non-semantic substrate with a fixed optimizer-memory budget is an original and potentially interesting approach to resource-efficient training. The use of minimalist binary embeddings to demonstrate representation learning is a creative and thought-provoking idea.

Weaknesses

W1. The empirical claims lack sufficient rigor. All controlled comparisons are based on single runs, with no statistical analysis of significance. The performance differences between the proposed method and baselines are small (the paper notes MMLU differences are often within a narrow band, see Fig. 5) and could fall within the expected variance of such experiments. While the authors focus on "stability patterns," this does not excuse the lack of rigor needed for making meaningful comparisons of competitiveness.

W2. The web-scale experiment is significantly confounded. The data mixture is changed between growth stages by interleaving instruction-tuning data. This makes it impossible to isolate the effect of the growth methodology from the effect of the data curriculum. The experiment also lacks a monolithic baseline, rendering the final MMLU score of 28.92% difficult to interpret in context.

W3  The downstream evaluation, while showing competitive results for the given model size, is limited in scope. The controlled study focuses on 248M-parameter models, where final MMLU scores are around 25-27%. While reasonable for this scale, the small performance differences between methods are hard to assess without multiple runs. The evaluation also lacks breadth, omitting key capabilities like generation or complex reasoning, which might be affected differently by the growth method.

W4 The paper's engagement with related work is very limited, citing only six references. It fails to discuss a large body of existing literature on progressive model growth and layer-wise training. This omission prevents a proper contextualization of the proposed method.

---

> ### Author Rebuttal · Authors · 2026-03-30
>
> Thank you for the thoughtful review. We agree that the current paper should be read more narrowly. It is not intended as a statistically rigorous claim of superiority over monolithic training, nor as a claim that this is the best general Transformer-growth recipe. Its intended claim is a feasibility/tradeoff claim: once useful learning can proceed above a fixed deterministic bottom interface (as motivated by the prior anonymous work cited in the paper), continued scaling need not remain monolithic, and depth can be increased while keeping the local trainable set / optimizer-state footprint bounded. The frozen 16-d binary token-ID substrate is important here because it is the strongest stress test of that fixed-interface premise.
>
> (1) Statistical analysis / competitiveness. We did not run multiple seeds for the controlled study. Under a fixed compute budget, we prioritized cross-scale replication (248M controlled study, 2.3B feasibility run, and the longer-horizon 16-layer / 68.9B-token run) over multi-seed repeats at one point. We agree this does not justify strong claims about statistically significant small differences, and we will narrow wording such as “competitive” accordingly. The 9-layer controlled study was intentionally protocol-matched: same 4B-token mixture, tokenizer, architecture, and shared training recipe across models. Therefore it should be interpreted as a regime comparison under a common protocol, not as each method at its individually optimized best. Notably, the monolithic frozen-UNICODE baseline already achieves lower final PPL than the constructive model (3.71 vs 4.12), so our claim is explicitly a tradeoff claim rather than a superiority claim.
>
> (2) Web-scale experiment / confounding / missing monolithic baseline. We agree that the 16-layer / 68.9B-token run is confounded for causal attribution because the stage-wise SFT mixture changes. It should not be interpreted as isolating the effect of growth alone. Its purpose in the paper is narrower: a long-horizon viability demonstration showing that continued learning remains possible even on a frozen 16-d binary token-ID substrate. In contrast, the cleanest comparison in the paper is the 9-layer controlled study, where all models use the same 4B-token mixture and shared protocol. A matched dense monolithic baseline was not included for the 16-layer run because it would require a different systems regime (e.g., full-model distributed training / optimizer-state sharding) than the fixed-local-budget regime studied here. We agree that such a baseline would be a useful upper bound, and that the current large-scale evidence should be read as feasibility evidence rather than a clean monolithic comparison.
>
> (3) Related work. We agree that the related-work coverage is too sparse and should have included recent depth-growth / incremental-training papers (e.g., On the Effectiveness of Incremental Training of Large Language Models, Curriculum-Guided Layer Scaling, progressive stacking / Net2Net / LiGO / stacking studies). The intended distinction is not simply that “growth exists,” but the specific regime studied here: fixed local trainable-parameter budget, freezing of all previously trained dense blocks, and LoRA used only as a budget-preserving global adjustment rather than dense full-model fine-tuning. We will expand this positioning and make the distinction clearer.
>
> We also agree that the downstream evaluation is limited in breadth. The paper evaluates selected tasks (MMLU, ARC, C-SENSE, SQuAD) to test whether useful performance survives under the fixed-interface / bounded-budget constraint; it is not intended as a comprehensive capability study. The absolute scores are modest because these models are constraint-driven and not benchmark-optimized endpoints. Under this narrower reading—as a fixed-local-budget feasibility/training-regime paper rather than a superiority claim—we hope the contribution is clearer.

---

> > ### Author Rebuttal · Reviewer_KiB9 · 2026-04-03
> >
> > not fully resolved

---

> > > ### Author Response · Authors · 2026-04-03
> > >
> > > Thank you again. We agree that, in its current form, the paper does not support statistically strong competitiveness claims and that the 68.9B-token run is confounded for causal attribution. The defensible reading is narrower: feasibility/tradeoff under a fixed active trainable-parameter budget, with the 9-layer common-protocol study as the cleanest comparison and the longer run as viability only. We also agree that the related-work coverage needs major expansion. We appreciate the candid assessment; it has helped clarify how the paper should be revised.

---

### Official Review · Reviewer_oPin · 2026-03-17

**Soundness:** 1
**Presentation:** 2
**Significance:** 3
**Originality:** 2
**Overall Recommendation:** 3
**Confidence:** 3

**Summary:**

This paper proposes a layer-wise growth scheme for Transformers that keeps the trainable-parameter / optimizer-memory budget approximately fixed as model depth increases. The method trains newly added layers while freezing earlier layers and regularly uses LoRA adapters across the existing stack as a budget-preserving global adjustment step. The experimental section includes a controlled 247.6M, 9-layer, 4B-token study against monolithic baselines and embedding ablations, a feasibility demonstration up to 2.3B parameters, and a 68.9B-token extended run with frozen 16-dimensional binary token-ID embeddings grown from 4 to 16 layers. I believe this paper is a distinct alternative to prior layer-wise growth approaches because it avoids full-model fine-tuning and keeps the optimizer-state footprint bounded throughout training.

**Compliance With Llm Reviewing Policy:**

Affirmed.

**Key Questions For Authors:**

Following the strengths and weaknesses section my questions are:
- Did the authors perform any learning-rate and/or batch-size tuning for the monolithic baseline, or otherwise verify that the baseline is near its best regime? This would materially affect how I interpret the controlled comparison.

- Could the authors clarify how they view this paper relative to recent depth-growth / incremental-training work such as On the Effectiveness of Incremental Training of Large Language Models and Curriculum-Guided Layer Scaling, and why these papers were not discussed in related work?

- How were the LoRA hyperparameters chosen (e.g. rank and target modules), and how was the number of layers updated in each round chosen for the 9-layer and 16-layer experiments? This is important for reproducibility and for understanding how sensitive the method is to these design choices.

- Could the authors clarify the evidence for the claimed interaction in Figure 5 between the growth procedure and the stable embedding substrate? My reading is that the figure suggests a strong main effect of the frozen UNICODE substrate under both monolithic and constructive training, rather than a demonstrated interaction. If I am missing something, clarification here would be helpful.

**Limitations:**

yes

**Strengths And Weaknesses:**

Strengths:

The paper explores an interesting and practically relevant idea of keeping the trainable-parameter / optimizer-state memory approximately fixed while growing model depth. The combination of staged depth growth with regular LoRA-based global adjustment is a reasonable and somewhat original variant of prior progressive-training ideas, since earlier works typically use global readjustment that increases optimizer-state memory beyond the fixed budget targeted here. The paper does include both a controlled comparison and larger feasibility-style experiments. The authors do show degradation relative to the baseline across perplexity and downstream tasks, but this tradeoff could still be acceptable if the optimizer-memory savings are compelling in settings where optimizer state is the dominant memory bottleneck. The larger scale experiments to show feasibility are useful as at larger scales is when the optimiser memory becomes a meaningful component of GPU memory compared to activations.

Weaknesses:

The only matched controlled comparison is at a modest scale (9 layers, 247.6M parameters, 4B tokens). Although the paper states that controlled comparisons use shared hyperparameters I do not see evidence that the monolithic baseline was independently tuned first. Without a baseline-specific hyperparameter search it is hard to know whether the reported differences reflect the method itself or a particular shared optimization setting. Also, the proposed constructive model reaches PPL 4.12 versus 3.71 for the monolithic frozen UNICODE baseline, an approximately 11% worse perplexity. This could be acceptable depending on the situation but I think that this is a main result that should be in the main paper with a convincing tradeoff argument to improve presentation as it was hard to read from the plots and I needed to find it in the appendix.

The larger scale experiments could be more convincing. The 2.3B model for the feasibility study seems unusually shallow at 6 layers. This and the extended 68.9B-token run are only reported for the proposed method without a matched monolithic baseline which would be very useful to see if the perplexity degradation changes as we scale to practical model sizes. Key LoRA details are also missing: the paper says rank and target modules were chosen to fit the budget, but does not clearly report them which hurts reproducibility.

I also think some of the paper’s interpretations are stronger than the evidence supports. The claim of an interaction between the growth procedure and a stable embedding substrate seems overstated. Figure 5 MMLU% seems consistent with a strong main effect of the frozen UNICODE substrate under both monolithic and constructive training, rather than a demonstrated interaction between the growth procedure and the presence of a stable embedding substrate.

The related-work discussion appears incomplete. The paper discusses older greedy layer-wise training and related modular/frozen-model ideas but I do not see discussion of more recent closely related depth-growth work such as On the Effectiveness of Incremental Training of Large Language Models or Curriculum-Guided Layer Scaling.

Useful additional baselines/ablations could include a baseline with a full model updating phase instead of lora to see the degradation lora produces as this seems like the main novelty of the paper. A baseline where we train a randomly chosen full layer under the same trainable-parameter budget and compare to the proposed newest top block only method could also be interesting. An ablation over growth chunk size / number of blocks added per stage could also show the resilience of the technique and optimise the difference in the perplexity compared to the baseline. Some careful accounting of the actual speed and memory gains/losses would help with demonstrating improved understanding.

Overall, I think the paper addresses a meaningful problem and has some originality in the exact training recipe, but the current empirical evidence is not strong enough to support the broader claims.

---

> ### Author Rebuttal · Authors · 2026-03-30
>
> Thank you for the careful and constructive review. We agree that the paper should be read more narrowly. The submission is not intended as the best general Transformer-growth recipe, nor as a claim of universal parity with monolithic training. Its question is narrower: once useful learning can proceed above a fixed deterministic bottom interface, can continued scaling also proceed without reopening the bottom of the network, while keeping the local trainable set / optimizer-state footprint bounded? Under this reading, the 9-layer study is the main controlled comparison, and the 2.3B and 16-layer runs are feasibility/viability demonstrations rather than matched monolithic comparisons.
>
> (1) Baseline tuning. We did not run a separate learning-rate / batch-size sweep for the monolithic 9-layer baseline, nor independently verify that it is at its own best regime. The controlled 9-layer study was intentionally protocol-matched: same 4B-token mixture, tokenizer, architecture, and shared training recipe across models. Therefore, the comparison should be interpreted as a regime comparison under a common protocol, not as each method at its individually optimized best. We agree this limits the strength of any absolute “competitiveness” claim and will narrow the wording accordingly. In fact, the monolithic frozen-UNICODE baseline already achieves lower final PPL (3.71 vs 4.12), so our claim is explicitly a tradeoff claim: at the same final 9-layer architecture, peak trainable parameters drop from 247.6M to 105.0M while retaining useful downstream performance. We also agree that this tradeoff should be made more explicit in the main text rather than being left hard to infer from the plots.
>
> (2) Relation to recent depth-growth / incremental-training work. We agree that the related-work section should have discussed recent papers such as On the Effectiveness of Incremental Training of Large Language Models and Curriculum-Guided Layer Scaling; this omission should be corrected. Our intended distinction is the regime studied here: all previously trained dense blocks remain frozen, the local trainable set is kept approximately fixed throughout, and LoRA is used only as a budget-preserving global adjustment rather than dense full-model fine-tuning. No stage trains the full dense model end-to-end. We will expand the related-work comparison accordingly.
>
> (3) LoRA hyperparameters and number of layers updated per round. There is no LoRA in the controlled 9-layer study. Growth chunk sizes were chosen mechanically by budget matching, not tuned for best perplexity: 3 blocks in the 9-layer study (105M newest blocks + LM head), 4 blocks in the 16-layer run (117.5M), and 1 block in the 2.3B run (738M). The 2.3B model is intentionally wide/shallow because, at d_model=4096, one newly trainable block + LM head already consumes almost the entire fixed local budget. In the 16-layer run, LoRA targeted q_proj, k_proj, v_proj, o_proj, mlp.net.0, and mlp.net.2, with lora_dropout=0.05, alpha=r, and bias="none". The rank r was chosen to match the same local trainable budget using r ≈ desired_trainable / Σ(in+out) over the matched linear modules, giving r=797 at 1–8 LoRA, r=531 at 1–12 LoRA, and r=398 at 1–16 LoRA. We agree that sensitivity ablations over chunk size / layer selection / LoRA design would be useful future work; in the current paper these choices were budget-driven rather than HPO-driven.
>
> (4) Claimed interaction in Figure 5. We agree that the word “interaction” was too strong. Figure 5 does not establish a formal statistical interaction. The evidence supports a narrower statement: constructive growth is compatible with a fixed substrate, and in this setup the constructive-trainable ablation underperforms its monolithic counterpart, suggesting that a stable bottom interface may help. We will revise this wording to avoid overstating the evidence. In particular, the frozen 16-d binary token-ID substrate is important here as the strongest stress test of the fixed-interface premise, not as a separate claim that this is the best overall growth recipe.
>
> On the larger-scale runs: we agree they are feasibility demonstrations rather than matched monolithic comparisons, and they should be read as such. A dense monolithic baseline at 2.3B in this setting would require a different systems regime (e.g., optimizer-state sharding / distributed full-model training) than the one studied here. Likewise, the 16-layer / 68.9B-token run is included as a long-horizon viability demonstration; because stage-wise SFT mixing is present, it should not be interpreted as a clean causal isolation of growth alone. We also agree that additional ablations such as a dense full-model updating phase, alternative layer-selection policies under the same budget, chunk-size sweeps, and explicit speed/memory accounting would strengthen a future revision. A dense full-model updating phase, however, would leave the fixed-local-budget regime that is central to this paper.

---

> > ### Author Rebuttal · Reviewer_oPin · 2026-04-02
> >
> > I believe the main focus of the paper is valid - evaluating the training of a model under a fixed parameter budget so that it can fit within a single H100 memory budget. This can naturally build on the already explored area of growing Transformers depth-wise.
> >
> > The scheme of adding a new block whilst keeping earlier layers frozen is not entirely novel, as it is present in "Progressively Stacking 2.0" (noted by Reviewer C1kq) for BERT, and similar schemes of freezing old blocks whilst adding and training new blocks for LLMs (not necessarily only the top block) appear in papers like "LESA" and "LLaMA Pro". The contribution of using LoRA to have global adjustment phases whilst staying under the budget is a nice novelty, given my understanding of the area, and the authors’ method also proposes random initialization compared to the G_stack approach of copying trained blocks that Reviewer C1kq pointed out. Given this rebuttal phase, I think the authors have better phrased the positioning of the paper, which I believe is still a valid direction to study and share results in.
> >
> > However, I do think that the initial related work was so severely limited that I am not even sure the reviewers were able to find all of the gaps during this review period. Furthermore, if I were a practitioner faced with this parameter-budget constraint, I would unfortunately still be confused about whether or not to proceed with training a model in this style, given the incomplete experimental results of the paper.
> >
> > 1) This 9-layer model was the only experiment that compared performance to vanilla training. It is completely fine, and expected, for the method to be worse than vanilla training, but I cannot reasonably evaluate the drop in performance I should expect from the method because the vanilla baseline is not tuned. This could simply be a particular learning-rate setting, for example, where vanilla training is poor and depth-wise growing appears not to damage the loss as much in comparison as it ordinarily might. I also would have preferred to see this experiment done with a standard pretraining dataset.
> >
> > An experiment with LoRA here, or just any comparison of the method with and without LoRA global adapter steps, seems critical to me, as that is a main novelty of the paper.
> >
> > 3) I understand that the authors position this larger-model experiment as a feasibility study. As other papers have already explored this progressive-growth topic, I think it is already established that this type of method would not completely collapse during training. I therefore still think it would be important, either in this section or another, to include some of the other baselines mentioned by other reviewers, in order to provide readers with some frame of reference for the performance of the method compared to other schemes, ideally at this larger model size.
> >
> > Reviewer o45H mentioned that 28.92% MMLU and 19.10% SQuAD EM are low, and the authors mentioned that these are only low compared to large modern LLMs, but there should at least be some evidence in the paper that a monolithic LLM of a similar scale would also get similarly low scores in order to validate the claim in Section 4.3: "In summary, this controlled study demonstrates that constructive growth on a frozen substrate is not only a viable training strategy and can be competitive with monolithic training on reasoning-heavy evaluations in this setup." This point was also raised by Reviewer KiB9, and the authors did not commit to conducting this baseline experiment.
> >
> > To conclude, I think the main aim of the study in the paper is fair, but the authors did not commit to addressing the experimental limitations raised by me and other reviewers. Also, the related work in the initial draft was severely limited, and I am not confident in the reviewers’ combined ability to find all the gaps during the rebuttal period. Therefore, I am maintaining my score.

---

> > > ### Author Response · Authors · 2026-04-03
> > >
> > > Thank you again. Your practitioner-oriented concern is well taken.
> > >
> > > If a practitioner can afford tuned dense monolithic training with sharding, that remains the natural stronger baseline. Our intended takeaway is narrower: when one explicitly wants to keep the active trainable set / optimizer-state footprint bounded at each stage and avoid dense full-model reopening, the paper provides a simple workable recipe and quantifies the tradeoff.
> > >
> > > In the controlled 9-layer study, that tradeoff is explicit:
> > > - peak active trainable parameters: 247.6M -> 105.0M
> > > - final PPL (monolithic frozen UNICODE vs constructive frozen): 3.71 -> 4.12
> > >
> > > So the claim should be read as “useful downstream performance survives under this memory-bounded regime,” not “this matches tuned monolithic training.” We agree the draft should have said this much more plainly and should have removed stronger competitiveness wording.
> > >
> > > We also agree with the main limitations you highlighted:
> > > 1. no baseline-specific HPO was performed, so the 9-layer result is a common-protocol regime comparison, not best-vs-best;
> > > 2. related work on recent growth / incremental-training papers needs substantial expansion;
> > > 3. comparisons against prior growth methods, and with/without LoRA in the controlled setting, are important missing baselines.
> > >
> > > That narrower practitioner framing is the one we would adopt in a revision, and we appreciate you spelling out why it matters.

---

### Decision · Program_Chairs · 2026-04-30

**Decision:**

Reject

**Comment:**

This paper proposes a layer-wise growth strategy for decoder-only Transformers, aiming to reduce the memory footprint of optimizer states during pretraining. Reviewers find the paper’s empirical claims unsubstantiated: its results rely on single runs without statistical analysis, and observed performance differences are minor. They also note the web-scale experiment is muddled by overlapping data mixture changes, the evaluation ignores broader capabilities, and the discussion of related work omits key recent works. The paper overstates evidence for interactions between growth and embedding and is undermined by missing baselines, ablations, and, e.g., LoRA details.

While its objective is sound and several ideas are interesting, it does not reach the ICML bar and I thus recommend its rejection.